# Mapping the zoonotic niche of Ebola virus disease in Africa

**David M Pigott[1][†], Nick Golding[1][†], Adrian Mylne[1], Zhi Huang[1], Andrew J Henry[1], Daniel J Weiss[1], Oliver J Brady[1], Moritz UG Kraemer[1], David L Smith[1,2], Catherine L Moyes[1], Samir Bhatt[1], Peter W Gething[1], Peter W Horby[3], Isaac I Bogoch[4,5], John S Brownstein[6,7], Sumiko R Mekaru[8], Andrew J Tatem[9,10,13], Kamran Khan[4,11], Simon I Hay[1,12]***

[1]Spatial Ecology and Epidemiology Group, Department of Zoology, University of Oxford, Oxford, United Kingdom; [2]Sanaria Institute for Global Health and Tropical Medicine, Rockville, United States; [3]Epidemic Diseases Research Group, Centre for Tropical Medicine and Global Health, University of Oxford, Oxford, United Kingdom; [4]Department of Medicine, Division of Infectious Diseases, University of Toronto, Toronto, Canada; [5]Divisions of Internal Medicine and Infectious Diseases, University Health Network, Toronto, Toronto, Canada; [6]Department of Pediatrics, Harvard Medical School, Boston, United States; [7]Children's Hospital Informatics Program, Boston Children's Hospital, Boston, United States; [8]Children's Hospital Informatics Program, Boston Children's Hospital, Boston, United States; [9]Department of Geography and Environment, University of Southampton, Southampton, United Kingdom; [10]Fogarty International Center, National Institutes of Health, Bethesda, United States; [11]Li Ka Shing Knowledge Institute, St. Michael's Hospital, Toronto, Canada; [12]Fogarty International Center, National Institutes of Health, Bethseda, United States; [13]Flowminder Foundation, Stockholm, Sweden

*For correspondence: simon.hay@zoo.ox.ac.uk

[†]These authors contributed equally to this work

**Abstract** Ebola virus disease (EVD) is a complex zoonosis that is highly virulent in humans. The largest recorded outbreak of EVD is ongoing in West Africa, outside of its previously reported and predicted niche. We assembled location data on all recorded zoonotic transmission to humans and Ebola virus infection in bats and primates (1976–2014). Using species distribution models, these occurrence data were paired with environmental covariates to predict a zoonotic transmission niche covering 22 countries across Central and West Africa. Vegetation, elevation, temperature, evapotranspiration, and suspected reservoir bat distributions define this relationship. At-risk areas are inhabited by 22 million people; however, the rarity of human outbreaks emphasises the very low probability of transmission to humans. Increasing population sizes and international connectivity by air since the first detection of EVD in 1976 suggest that the dynamics of human-to-human secondary transmission in contemporary outbreaks will be very different to those of the past.

## Introduction

Ebola viruses have for the last forty years been responsible for a number of outbreaks of Ebola virus disease (EVD) in humans (*Pattyn et al., 1977*), with high case fatality rates typically around 60–70%, but potentially reaching as high as 90% (*Feldmann and Geisbert, 2011*). The most recent outbreak began in Guinea in December 2013 (*Baize et al., 2014*; *Bausch and Schwarz, 2014*) and has subsequently spread to Liberia, Sierra Leone and Nigeria (*ECDC, 2014*). The unprecedented size and scale of this ongoing outbreak has the potential to destabilise already fragile economies and healthcare

**eLife digest** Since the first outbreaks of Ebola virus disease in 1976, there have been numerous other outbreaks in humans across Africa with fatality rates ranging from 50% to 90%. Humans can become infected with the Ebola virus after direct contact with blood or bodily fluids from an infected person or animal. The virus also infects and kills other primates—such as chimpanzees or gorillas—though Old World fruit bats are suspected to be the most likely carriers of the virus in the wild.

The largest recorded outbreak of Ebola virus disease is ongoing in West Africa: more people have been infected in this current outbreak than in all previous outbreaks combined. The current outbreak is also the first to occur in West Africa—which is outside the previously known range of the Ebola virus.

Pigott et al. have now updated predictions about where in Africa wild animals may harbour the virus and where the transmission of the virus from these animals to humans is possible. As such, the map identifies the regions that are most at risk of a future Ebola outbreak. The data behind these new maps include the locations of all recorded primary cases of Ebola in human populations—the 'index' cases—many of which have been linked to animal sources. The data also include the locations of recorded cases of Ebola virus infections in wild bats and primates from the last forty years. The maps, which were modelled using more flexible methods than previous predictions, also include new information—collected using satellites—about environmental factors and new predictions of the range of wild fruit bats.

Pigott et al. report that the transmission of Ebola virus from animals to humans is possible in 22 countries across Central and West Africa—and that 22 million people live in the areas at risk. However, outbreaks in human populations are rare and the likelihood of a human getting the disease from an infected animal still remains very low. The updated map does not include data about how infections spread from one person to another, so the next challenge is to use existing data on human-to-human transmission to better understand the likely size and extent of current and future outbreaks. As more people live in, and travel to and from, the at-risk regions than ever before, Pigott et al. note that new outbreaks of Ebola virus disease are likely to be very different to those of the past.

systems (*Fauci, 2014*), and fears of international spread of a Category A Priority Pathogen (*NIH, 2014*) have made this a massive focus for international public health (*Chan, 2014*). This has led to the current outbreak being declared a Public Health Emergency of International Concern on the 8 August 2014 (*Briand et al., 2014*; *Gostin et al., 2014*; *WHO, 2014d*).

The *Filoviridae*, of which *Ebolavirus* is a constituent genus, belong to the order *Mononegavirales*. Two other genera complete the family: *Marburgvirus*, itself responsible for a number of outbreaks of haemorrhagic fever across Africa (*Gear et al., 1975*; *Conrad et al., 1978*; *Smith et al., 1982*; *Towner et al., 2006*) and *Cuevavirus*, recently isolated from bats in northern Spain (*Negredo et al., 2011*). Five species of *Ebolavirus* have been isolated to date (*Kuhn et al., 2010*; *King et al., 2012*); the earliest recognised outbreaks of EVD were reported in Zaire (now the Democratic Republic of the Congo [DRC]) and Sudan in 1976 (*International Commission, 1978*; *WHO International Study Team, 1978*). The causative viruses were isolated (*Pattyn et al., 1977*) and later identified to be distinct species, *Zaire ebolavirus* (EBOV) and *Sudan ebolavirus* (SUDV). A third species of *Ebolavirus*, *Reston ebolavirus*, was isolated from Cynomologus monkeys imported from the Philippines to a facility in the United States, where they experienced severe haemorrhaging (*Jahrling et al., 1990*). Whilst serological evidence of infection with this species has been reported in individuals in the Philippines (*Miranda et al., 1991*), no pathogenicity has been reported beyond primates and porcids (*Barrette et al., 2009*; *Feldmann and Geisbert, 2011*). In 1994 a fourth species, *Tai Forest ebolavirus* was isolated from a veterinarian who had autopsied a chimpanzee in Côte d'Ivoire (*Le Guenno et al., 1995*), though the virus has not been detected subsequently. The final species, *Bundibugyo ebolavirus*, was responsible for an outbreak of EVD in Uganda in 2007 (*Towner et al., 2008*), as well as a more recent outbreak in the DRC (*WHO, 2012b*).

Initial analysis suggested that the viruses isolated from the current outbreak, originating in Guinea, formed a separate clade within the five *Ebolavirus* species (*Baize et al., 2014*). Subsequent re-analysis

of the same sequences however, indicated that these isolates instead nest within the *Zaire ebolavirus* lineage (*Dudas and Rambaut, 2014*), and diverged from Central Africa strains approximately ten years ago (*Gire et al., 2014*).

Which reservoir species are responsible for maintaining Ebola transmission between outbreaks is not well understood (*Peterson et al., 2004b*), but over the last decade significant progress has been made in narrowing down the list of likely hosts (*Peterson et al., 2007*) (*Figure 1*). Primates have long been known to harbour filoviral infections, with the first Marburg strains identified in African green monkeys in 1967 (*Siegert et al., 1967*; *Beer et al., 1999*). Significant mortality has also been reported in wild primate populations across Africa, most notably in gorilla (*Gorilla gorilla*) and chimpanzee (*Pan troglodytes*) populations (*Formenty et al., 1999*; *Rouquet et al., 2005*; *Bermejo et al., 2006*). The high case fatality rates recorded in the great apes combined with their declining populations and limited geographical range, indicate they are likely dead-end hosts for the virus and not reservoir species (*Groseth et al., 2007*). A large survey of small mammals in and around Gabon identified three species of bats which were infected with Ebola viruses—*Hypsignathus monstrosus*, *Epomops franqueti* and *Myonycetris torquata* (*Leroy et al., 2005*). Subsequent serological surveys (*Pourrut et al., 2009*; *Hayman et al., 2010*) and evidence linking the potential source of human outbreaks to bats (*Leroy et al., 2009*) lend support to the hypothesis of a bat reservoir. This, coupled with repeated detection of *Marburgvirus* in the fruit bat *Rousettus aegypticus* (*Towner et al., 2009*) and the only isolations of *Cuevavirus* also from bats (specifically *Llovia virus* [*Negredo et al., 2011*]), all support the suspicion that Chiroptera play an important role in the natural life-cycle of the filoviruses.

Humans represent a dead-end host for the virus, with only stuttering chains of transmission reported between humans in the majority of previous outbreaks (*Chowell et al., 2004*; *Legrand et al., 2007*) and no indication that humans can reintroduce the virus back into reservoir species (*Karesh et al., 2012*). The incubation period in humans ranges from two days to three weeks, after which a variety of clinical symptoms arise, affecting multiple organs of the body. At the peak of illness, haemorrhaging shock and widespread tissue damage can occur and can eventually lead to death within 6–16 days (*Feldmann and Geisbert, 2011*). Human-to-human transmission is mainly through direct unprotected contact with infected individuals and cadavers, with infectious particles detected in a number of different body fluids (*Feldmann and Geisbert, 2011*). The typical outbreak profile is defined by an index individual that has recently come into contact with the blood of another mammal through either hunting or the butchering of animal carcasses (*Pourrut et al., 2005*). Whilst it has been difficult to identify the zoonotic source for the index cases of some outbreaks, a recurring theme of hunting and handling bushmeat is suspected (*Table 1*; *Boumandouki et al., 2005*; *Nkoghe et al., 2005, 2011*; *Leroy et al., 2009*). For some outbreaks, including the most recent, the initial source of zoonotic transmission has not been identified. In subsequent human-to-human transmission, the highest risk activities are those that bring humans into close contact with infected individuals. These include medical settings where insufficient infection control precautions have been taken, as well as home care and funeral preparations carried out by families or close friends (*Baron et al., 1983*; *Georges et al., 1999*; *Boumandouki et al., 2005*). As the conditions required for transmission are culturally and contextually dependent, opportunities for sustained transmission are highly heterogeneously distributed. Typically, chains of infection do not exceed three or four sequential transmission events, although occasionally (and particularly in

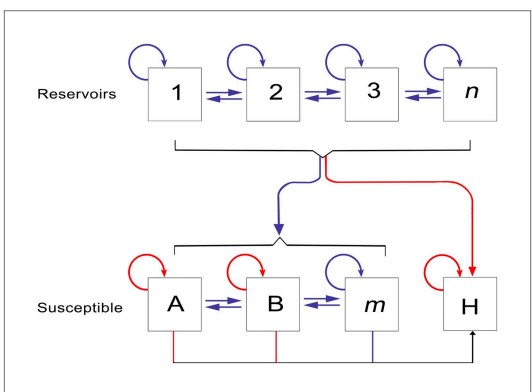

**Figure 1**. The epidemiology of Ebola virus transmission in Africa. Of the suspected reservoir species, 1, 2 and 3 represent the three bat species from which Ebola virus has been isolated (*Hypsignathus monstrosus*, *Myonycteris torquata* and *Epomops franqueti*) and *n* represents unknown reservoirs of the disease yet to be discovered. Of the susceptible species, A represents *Pan troglodytes*, B *Gorilla gorilla* and *m* represents other organisms susceptible to the disease, such as duikers. H represents humans. Blue arrows indicate unknown transmission cycles or infection routes and red arrow routes have been confirmed or are suspected. Adapted from *Groseth et al. (2007)*.

**Table 1.** Locations of outbreaks of Ebola virus disease in humans

| Outbreak | Countries | Date range | Location | Species | Reference |
|---|---|---|---|---|---|
| 1 | South Sudan | Jun–Nov 1976 | Nzara | SUDV | (*WHO International Study Team, 1978*) |
| 2 | DRC | Sep–Oct 1976 | Yambuku | EBOV | (*International Commission, 1978*) |
| 3 | DRC | Jun 1977 | Bonduni | EBOV | (*Heymann et al., 1980*) |
| 4 | South Sudan | Jul–Oct 1979 | Nzara | SUDV | (*Baron et al., 1983*) |
| 5 | Côte d'Ivoire | Nov 1994 | Tai Forest | TAFV | (*Le Guenno et al., 1995*; *Formenty et al., 1999*) |
| 6 | Gabon | Nov 1994–Feb 1995 | Mekouka and Andock mining camps | EBOV | (*Amblard et al., 1997*; *Georges et al., 1999*; *Milleliri et al., 2004*) |
| 7 | DRC | Jan–Jul 1995 | Mwembe Forest | EBOV | (*Muyembe and Kipasa, 1995*; *Khan et al., 1999*) |
| 8 | Gabon | Jan–Mar 1996 | Mayibout 2 | EBOV | (*Georges et al., 1999*; *Milleliri et al., 2004*) |
| 9 | Gabon | Jul 1996–Jan 1997 | Booue | EBOV | (*Georges et al., 1999*; *Milleliri et al., 2004*) |
| 10 | Uganda | Oct 2000–Feb 2001 | Rwot-Obillo | SUDV | (*WHO, 2001*; *Okware et al., 2002*; *Lamunu et al., 2004*) |
| 11 | Gabon & ROC | Oct 2001–Mar 2002 | Memdemba Entsiami, Abolo and Ambomi | EBOV | (WHO, 2003; *Milleliri et al., 2004*; *Nkoghe et al., 2005*; *Pourrut et al., 2005*) |
| | | | Ekata | | |
| | | | Oloba | | |
| | | | Etakangaye | | |
| | | | Grand Etoumbi | | |
| 12 | ROC | Dec 2002–Apr 2003 | Yembelangoye | EBOV | (WHO, 2003; *Pourrut et al., 2005*) |
| | | | Nearby hunting camp | | |
| | | | Mvoula | | |
| 13 | ROC | Oct–Dec 2003 | Mbandza | EBOV | (*Boumandouki et al., 2005*) |
| 14 | South Sudan | Apr–Jun 2004 | Forests bordering Yambio | SUDV | (*WHO, 2005*; *Onyango et al., 2007*) |
| 15 | ROC | Apr–May 2005 | Odzala National Park | EBOV | (*Nkoghe et al., 2011*) |
| 16 | DRC | May–Nov 2007 | Mombo Mounene 2 market | EBOV | (*Leroy et al., 2009*) |
| 17 | Uganda | Aug–Dec 2007 | Kabango | BDBV | (*Towner et al., 2008*; *MacNeil et al., 2010*; *Wamala et al., 2010*) |
| 18 | DRC | Nov 2008–Feb 2009 | Luebo | EBOV | (*Grard et al., 2011*) |
| 19 | Uganda | May 2011 | Nakisamata | SUDV | (*Shoemaker et al., 2012*) |
| 20 | DRC | July–Nov 2012 | Isiro | BDBV | (*CDC, 2014*; *WHO, 2012b*) |
| 21 | Uganda | July–Oct 2012 | Nyanswiga | SUDV | (*CDC, 2014*; *WHO, 2012a*) |
| 22 | Uganda | Nov 2012–Jan 2013 | Luwero District | SUDV | (*WHO, 2012c*; *CDC, 2014*) |
| 23 | Guinea | Dec 2013 - | Meliandou | EBOV | (*Baize et al., 2014*; *Bausch and Schwarz, 2014*) |

DRC = Democratic Republic of the Congo, ROC = Republic of Congo.

the early stages of infection) a single individual may be responsible for directly infecting a large number of others (*Brady et al., 2014*). In the outbreak in Gabon in 1996, a single person was responsible for infecting ten other individuals (*Milleliri et al., 2004*) whilst in the 1995 outbreak in the DRC, thirty five cases resulted from one individual (*Khan et al., 1999*). Secondary transmission can be restricted by effective case detection and isolation measures (*Shoemaker et al., 2012*; *WHO, 2014c*). Where this cannot be achieved, either due to a lack of infrastructure, poor understanding of the disease, or distrust of medical practices, secondary cases can continue to occur (*Khan et al., 1999*; *Larkin, 2003*; *Hewlett et al., 2005*). As the number of infections grows, the ability of healthcare systems to control the further spread diminishes and the risk of a large outbreak increases.

The recent outbreak in Guinea and surrounding countries indicate that the previous paradigm for Ebola outbreaks is shifting (*Briand et al., 2014*; *Chan, 2014*). The last 40 years of EVD outbreaks were accompanied by considerable changes in demographic patterns throughout Africa. There has been a large increase in population size coupled with increasing urbanisation (*Cohen, 2004*; *Seto et al., 2012*; *Linard et al., 2013*). African populations have also become better connected internally and internationally (*Linard et al., 2012*; *Huang and Tatem, 2013*). Only recently have we begun to understand the dynamic nature of these travel patterns (*Garcia et al., 2014*; *Gonzalez et al., 2008*; *Simini et al., 2012*; *Wesolowski et al., 2013*, *2012*) which have been clearly demonstrated to influence disease transmission over different temporal and spatial scales (*Hufnagel et al., 2004*; *Yang et al., 2008*; *Stoddard et al., 2009*; *Talbi et al., 2010*; *Brockmann and Helbing, 2013*; *Pindolia et al., 2014*). Changes in land use and penetration into previously remote areas of rainforest bring humans into contact with potential new reservoirs (*Daszak, 2000*), while changes in human mobility and connectivity will likely have profound impacts on the dispersion of Ebola cases during outbreaks. These conditions are thought to have a major role in setting the stage for the current outbreak.

This paper aims to define the areas suitable for zoonotic transmission of *Ebolavirus* (i.e., those routes defined in *Figure 1* excluding human-to-human transmission) through species distribution modelling techniques. The fundamental niche of a species can be conceptualised as the confluence of environmental conditions that support its presence in a particular location (*Franklin, 2009*). Species distribution models quantitatively describe this niche based on known occurrence records of the organism and their associated environmental conditions, enabling predictions of the likely geographic distribution of the species in other regions (*Elith and Leathwick, 2009*). The era of satellites and geographical information systems has made high resolution global data on environmental conditions increasingly available (*Hay et al., 2006*; *Weiss et al., 2014b*). Species distribution modelling using flexible machine learning approaches have been successfully applied to map the global distributions of disease vectors (*Sinka et al., 2012*) and pathogens such as dengue (*Bhatt et al., 2013*), influenza (*Gilbert et al., 2014*) and leishmaniasis (*Pigott et al., 2014*).

Previous studies applied the GARP (Genetic Algorithm for Rule-set Production) species distribution modelling approach (*Stockwell and Peters, 1999*) to the locations of 12 Ebola outbreaks in humans between 1976 and 2002 to map the likely distribution of Ebola viruses (*Peterson et al., 2004a*) and as a mechanism to identify potential reservoir hosts (*Peterson et al., 2004b*; *Peterson et al., 2007*). Here we update and improve the maps of the zoonotic transmission niche of EVD by: (i) incorporating more recent outbreak data from outside the formerly predicted niche of EVD; (ii) integrating for the first time data on outbreaks in primates and the occurrence of the virus in the suspected Old World fruit bat (OWFB) reservoirs; (iii) using new satellite-derived information on bespoke environmental covariates from Africa, including new distribution maps of the OWFB; and (iv) using new increasingly flexible niche mapping techniques in the modelling framework. To elucidate the relevance of these maps for transmission, we have also calculated the population at risk of primary spillover outbreaks from the zoonotic niche of EVD in Africa, and we investigated the changing nature of the populations within this niche.

## Results

### Reported EVD outbreaks

In total, 23 outbreaks of Ebola virus were identified in humans across Africa, consisting of a hypothesised 30 independent primary infection events (*Table 1*; *Figure 2*). These outbreaks span the last 40 years from the first outbreaks in 1976 to the five outbreaks that have occurred since 2010 (*Table 1*). The locations of the index cases span from West Africa, with the most westerly outbreak ongoing in Guinea, to Gabon, the Republic of Congo (ROC), the DRC, South Sudan and Uganda. Before December

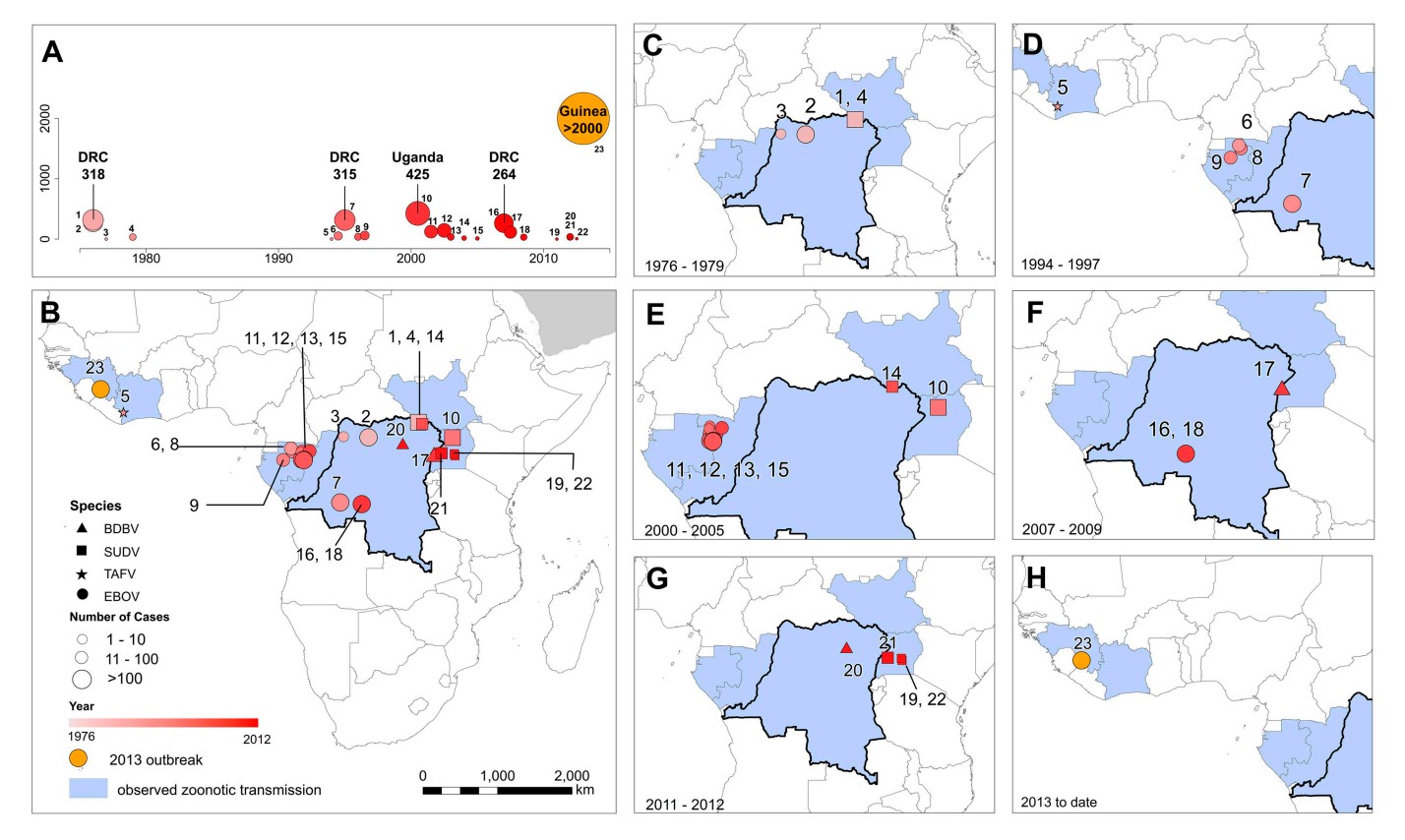

**Figure 2**. The locations of Ebola virus disease outbreaks in humans in Africa. (**A**) Illustrates the 23 reported outbreaks of Ebola virus disease through time, with the area of each circle and its position along the y-axis representing the number of cases. The onset year is represented by the colour as per (**B**). (**B**) Shows a map of the index cases for each of these outbreaks. (**C–H**) Show these outbreaks over a series of time periods. Numbers refer to outbreaks as listed in *Table 1*. In (**B–H**) the species of Ebola virus responsible for the outbreak is illustrated by the symbol shape, the number of resulting cases and onset date by symbol colour. The most recent outbreak (#23) is indicated in orange. Countries in which zoonotic transmission to humans has been reported or is assumed to have occurred are coloured in blue. In each map the Democratic Republic of Congo is outlined for reference.

2013, a total of 2322 cases had occurred from *Ebolavirus* infections, a number already overtaken by the likely underreported current case count of the ongoing outbreak >2250 (*WHO, 2014a*) (*Figure 2A*). Of the four viruses circulating in Africa, *Zaire ebolavirus* has been responsible for the most outbreaks (13), followed by *Sudan ebolavirus* (7) and *Bundigbuyo ebolavirus* with just two outbreaks in 2007/8 and 2012. Tai Forest has caused one confirmed infection in humans, from which the patient recovered (*Le Guenno et al., 1995*; *Formenty et al., 1999*). Although outbreaks have been reported since 1976, there was an absence of reported outbreaks in humans for 15 years between 1979 and 1994 (although antibodies in humans were identified over the period [*Kuhn, 2008*]) and the frequency of outbreaks has increased substantially post 2000 (*Figure 2A*).

### Reported Ebola virus infections in animals

A total of 51 surveyed locations reporting infections in animals were identified in the literature since the discovery of the disease (*Table 2*; *Figure 3*). These comprised 17 infections in gorillas (*Gorilla gorilla*), nine infections in chimpanzees (*Pan troglodytes*), 18 in OWFB and 2 in duikers (*Cephalophus* spp.). A large proportion of the great ape cases originated from the ROC/Gabon border, coinciding with the main known distributions of both chimpanzees and gorillas (*Petter and Desbordes, 2013*) and representing a period of well-documented great ape Ebola outbreaks in and around the Lossi Animal Sanctuary (*Rouquet et al., 2005*; *Bermejo et al., 2006*; *Walsh et al., 2009*). All animal isolations of Ebola viruses have come from countries that have also reported index cases of human outbreaks, with the exception of several seropositive bats from a survey in southern Ghana.

**Table 2.** Locations of reported infections with Ebola virus in animals

| Site | Country | Date range | Location | Species | Diagnosis | Reference |
|---|---|---|---|---|---|---|
| 1 | Côte d'Ivoire | Oct–Nov 1994 | Tai Forest | Chimpanzee | Serology | (*Formenty et al., 1999*) |
| 2 | Gabon | Jan 1996 | Mayiboth 2 | Chimpanzee | PCR | (*Lahm et al., 2007*) |
| 3 | Gabon | Jul 1996 | Near Booue | Chimpanzee | Serology | (*Georges-Courbot et al., 1997*) |
| 4 | Gabon | Sept 1996 | Lope National Park | Chimpanzee | PCR | (*Lahm et al., 2007*) |
| 5 | Gabon & ROC | Aug 2001 | Mendemba/Lossi Animal Sanctuary | Chimpanzee | PCR | (*Lahm et al., 2007*) |
| 6 | Gabon & ROC | Aug 2001 | Mendemba/Lossi Animal Sanctuary | Gorilla | PCR | (*Lahm et al., 2007*) |
| 7 | Gabon & ROC | Aug 2001 | Mendemba/Lossi Animal Sanctuary | *Cephalophus dorsalis* | PCR | (*Lahm et al., 2007*) |
| 8 | Gabon | Nov 2001 | Zadie | Gorilla | PCR | (*Rouquet et al., 2005*) |
| 9 | Gabon | Nov 2001 | Ekata | Gorilla | PCR | (*Wittmann et al., 2007*) |
| 10 | Gabon | Dec 2001 | Medemba and neighbouring villages | Chimpanzee and Gorilla | PCR | (*Leroy et al., 2002*) |
| 11 | Gabon | Feb 2002 | Zadie | Gorilla | PCR | (*Rouquet et al., 2005*) |
| 12 | Gabon | Feb 2002 | Ekata | Various bat species | Serology | (*Leroy et al., 2005*) |
| 13 | Gabon | Mar 2002 | Zadie | Gorilla | PCR | (*Rouquet et al., 2005*) |
| 14 | Gabon | Mar 2002 | Grand Etoumbi | Gorilla | PCR | (*Wittmann et al., 2007*) |
| 15 | Gabon | Apr 2002 | Ekata | Gorilla | PCR | (*Wittmann et al., 2007*) |
| 16 | ROC | May 2002 | Oloba | Chimpanzee | PCR | (*Lahm et al., 2007*) |
| 17 | ROC | Dec 2002 | Lossi Animal Sanctuary | Gorilla | PCR | (*Rouquet et al., 2005*) |
| 18 | ROC | Dec 2002 | Lossi Animal Sanctuary | Gorilla | PCR | (*Rouquet et al., 2005*) |
| 19 | ROC | Dec 2002 | Lossi Animal Sanctuary | Chimpanzee | Serology | (*Rouquet et al., 2005*) |
| 20 | ROC | Dec 2002 | Lossi Animal Sanctuary | Gorilla | PCR | (*Rouquet et al., 2005*) |
| 21 | ROC | Dec 2002 | Lossi Animal Sanctuary | Gorilla | PCR | (*Rouquet et al., 2005*) |
| 22 | ROC | Dec 2002 | Lossi Animal Sanctuary | *Cephalophus* spp. | PCR | (*Rouquet et al., 2005*) |
| 23 | Gabon | Feb 2003 | Mbomo | Various bat species | PCR | (*Leroy et al., 2005*) |
| 24 | ROC | Feb 2003 | Lossi Animal Sanctuary | Gorilla | Serology | (*Rouquet et al., 2005*) |
| 25 | Gabon | Feb 2003 | Lossi Animal Sanctuary | Chimpanzee | PCR | (*Wittmann et al., 2007*) |
| 26 | Gabon | Jun 2003 | Mbomo | Various bat species | PCR and serology | (*Leroy et al., 2005*) |
| 27 | ROC | Jun 2003 | Near Mbomo and Ozala National Park | *Epomops franqueti* | Serology | (*Pourrut et al., 2009*) |
| 28 | ROC | Jun 2003 | Near Mbomo and Ozala National Park | *Hypsignathus monstrosus* | Serology | (*Pourrut et al., 2009*) |
| 29 | ROC | Jun 2003 | Near Mbomo and Ozala National Park | *Myonycteris torquata* | Serology | (*Pourrut et al., 2009*) |

*Table 2. Continued on next page*

*Table 2. Continued*

| Site | Country | Date range | Location | Species | Diagnosis | Reference |
|------|---------|-----------|----------|---------|-----------|-----------|
| 30 | ROC | Jun 2003 | Mbanza | Gorilla | PCR | (*Rouquet et al., 2005*) |
| 31 | ROC | Jan–Jun 2004 | Lokoué | Gorilla | Reported | (*Caillaud et al., 2006*) |
| 32 | ROC | May 2004 | Lokoué | Gorilla | PCR | (*Wittmann et al., 2007*) |
| 33 | Gabon | Feb 2005 | Near Franceville | *Epomops franqueti* | Serology | (*Pourrut et al., 2009*) |
| 34 | Gabon | Feb 2005 | Near Franceville | *Myonycteris torquata* | Serology | (*Pourrut et al., 2009*) |
| 35 | Gabon | Apr 2005 | Near Lambarene | *Epomops franqueti* and *Hypsignathus monstrosus* | Serology | (*Pourrut et al., 2007*) |
| 36 | ROC | May 2005 | Near Mbomo and Ozala National Park | *Epomops franqueti* | Serology | (*Pourrut et al., 2009*) |
| 37 | ROC | May 2005 | Near Mbomo and Ozala National Park | *Hypsignathus monstrosus* | Serology | (*Pourrut et al., 2009*) |
| 38 | ROC | May 2005 | Near Mbomo and Ozala National Park | *Myonycteris torquata* | Serology | (*Pourrut et al., 2009*) |
| 39 | ROC | Jun 2005 | Odzala National Park | Gorilla | PCR | (*Wittmann et al., 2007*) |
| 40 | Gabon | Feb 2006 | Near Tchibanga | Various bat species | Serology | (*Pourrut et al., 2009*) |
| 41 | ROC | May 2006 | Near Mbomo and Ozala National Park | *Epomops franqueti* | Serology | (*Pourrut et al., 2009*) |
| 42 | ROC | May 2006 | Near Mbomo and Ozala National Park | *Hypsignathus monstrosus* | Serology | (*Pourrut et al., 2009*) |
| 43 | ROC | May 2006 | Near Mbomo and Ozala National Park | *Myonycteris torquata* | Serology | (*Pourrut et al., 2009*) |
| 44 | Gabon | Oct 2006 | Near Franceville | *Epomops franqueti* | Serology | (*Pourrut et al., 2009*) |
| 45 | Ghana | May 2007 | Sagyimase | *Epomops franqueti* | Serology | (*Hayman et al., 2012*) |
| 46 | Ghana | May 2007 | Sagyimase | *Hypsignathus monstrosus* | Serology | (*Hayman et al., 2012*) |
| 47 | Ghana | May 2007 | Adoagyir | *Epomophorus gambianus* | Serology | (*Hayman et al., 2012*) |
| 48 | Ghana | May 2007 | Adoagyir | *Epomops franqueti* | Serology | (*Hayman et al., 2012*) |
| 49 | Ghana | Jun 2007 | Oyibi | *Epomophorus gambianus* | Serology | (*Hayman et al., 2012*) |
| 50 | Ghana | Jan 2008 | Accra | *Eidolon helvum* | Serology | (*Hayman et al., 2010*) |
| 51 | Gabon | Mar 2008 | Near Franceville | *Epomops franqueti* | Serology | (*Pourrut et al., 2009*) |

ROC = Republic of Congo.

## Predicted distribution of suspected reservoir species of bats

Three species of bats, *Hypsignathus monstrosus*, *Myonycteris torquata* and *Epomops franqueti*, were identified as the most likely candidates to be reservoir species for Ebola viruses due to high

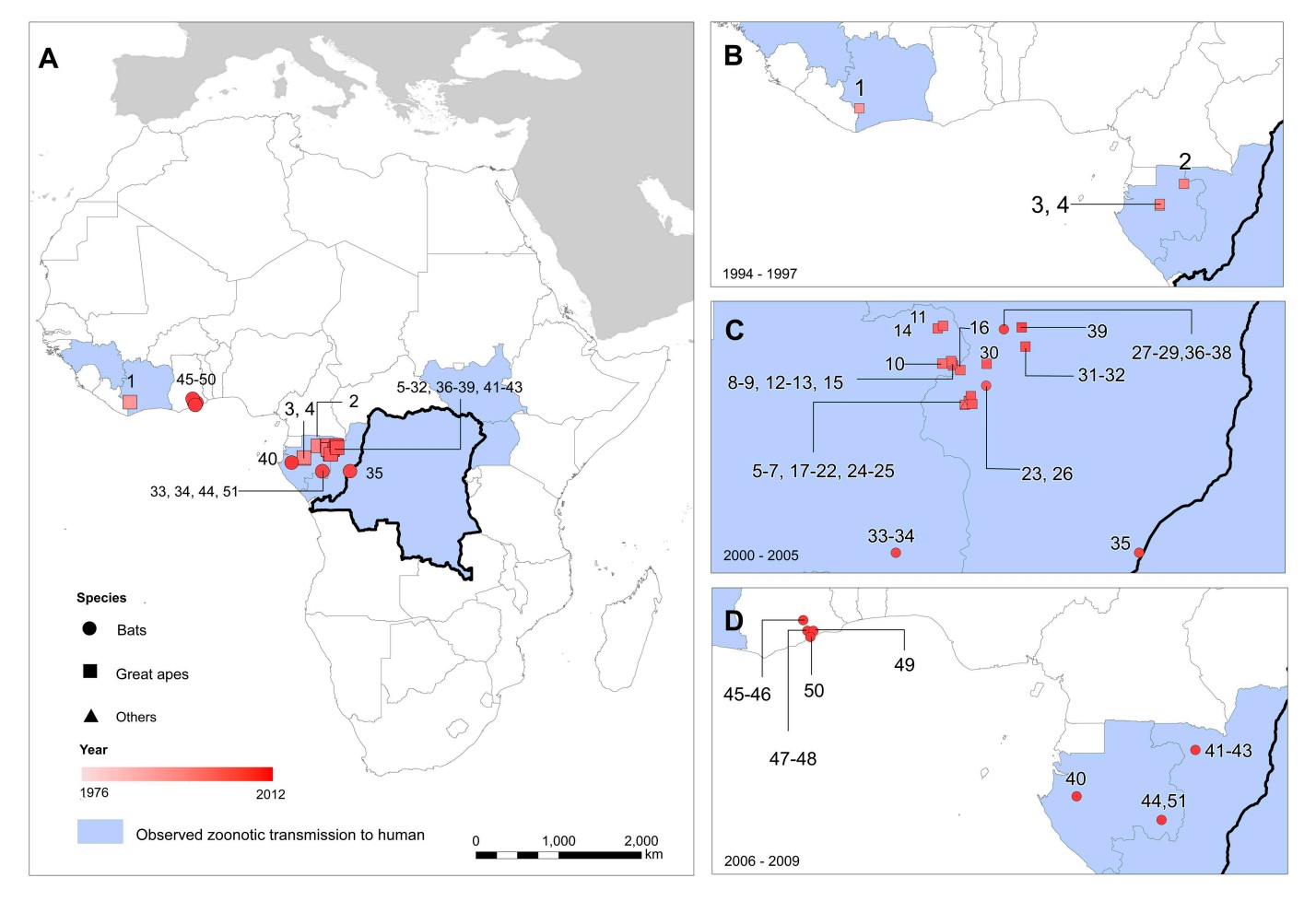

**Figure 3**. The locations of reported Ebola virus infection in animals in Africa. (**A**) Shows the locations of reported Ebola virus infection in animals. (**B**–**D**) Show these records in animals over three different time periods. Numbers refer to records as listed in *Table 2*. In all panels, the species in which infection was detected is given by symbol shape and the year recorded by symbol colour. Blue countries represent locations where zoonotic transmission to humans has been reported or is assumed to have occurred. In each map the Democratic Republic of Congo is outlined for reference.

seroprevalence and the isolation of RNA closely related to *Zaire ebolavirus* (*Leroy et al., 2005*; *Olival and Hayman, 2014*). In total, 239 locations were identified from the Global Biodiversity Information Facility (GBIF) (*GBIF, 2014*): 67 for *H. monstrosus* (*Figure 4A*), 52 for *M. torquata* (*Figure 4B*) and 120 for *E. franqueti* (*Figure 4C*). Distribution models for all three species demonstrated predictive skill (indicated by an area under the curve (AUC) greater than 0.5) as follows: *H. monstrosus* AUC 0.63 ± 0.04; *M. torquata* AUC = 0.59 ± 0.04; *E. franqueti* AUC = 0.58 ± 0.03, n = 50 submodels for all three species. In addition, each species was broadly predicted within its considered expert opinion range (*Figure 4A–C*) (*Schipper et al., 2008*). The marginal effect plots (not shown) were strongly influenced by land surface temperature (LST) and vegetation (as measured by the enhanced vegetation index [EVI]). The predicted combined distribution of these species (*Figure 4D*), covers West and Central Africa, specifically the moist forests of the northeastern, western and central Congo basin, and Guinea, as well as the Congolian coastal forest ecoregions (*WWF, 2014*).

## Predicted environmental suitability for zoonotic transmission of Ebola

The predicted environmental niche for zoonotic transmission of EVD is shown in *Figure 5*. All countries with observed index cases of EVD (n = 7, hereafter Set 1) have areas of the highest environmental suitability (see list in *Table 1*). In addition, areas of high environmental suitability for zoonotic

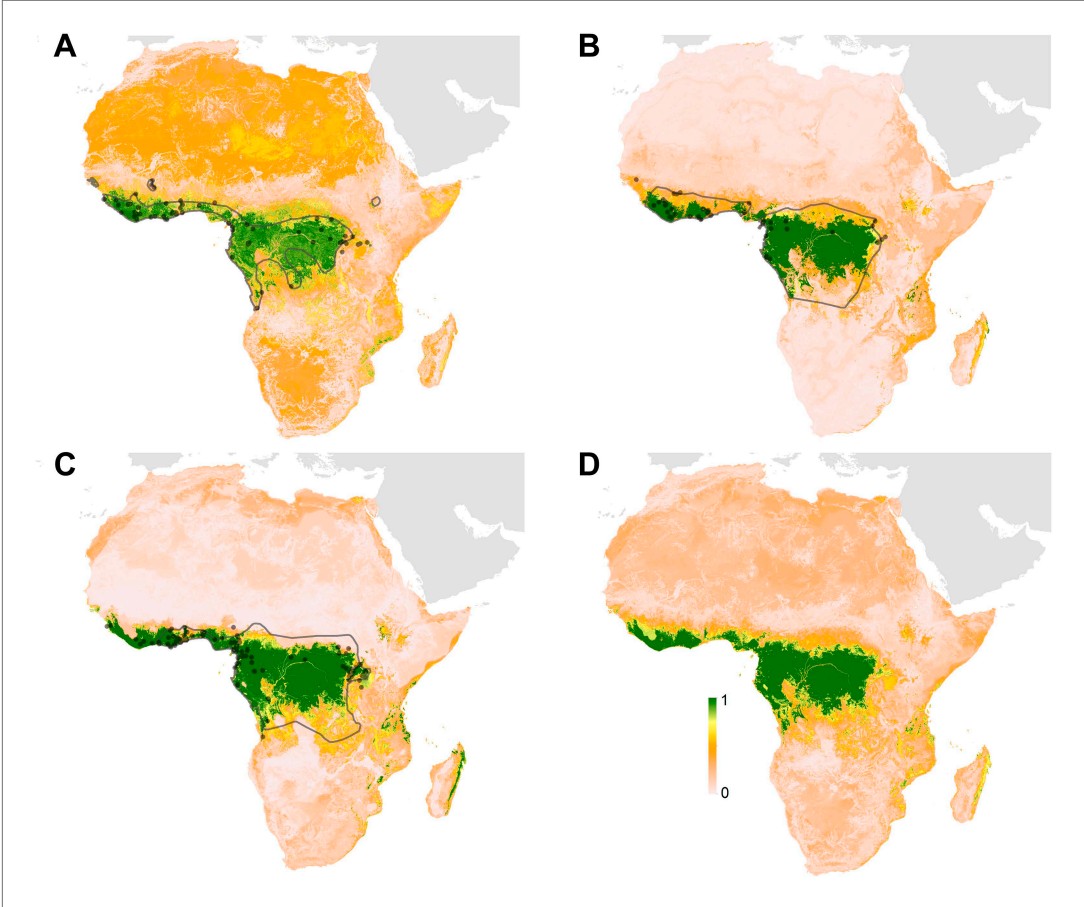

**Figure 4**. Predicted geographical distribution of the three species of Megachiroptera suspected to reservoir Ebola virus. (**A**) Shows the distribution of the hammer-headed bat (*Hypsignathus monstrosus*), (**B**) The little collared fruit bat (*Myonycteris torquata*) and (**C**) Franquet's epauletted fruit bat (*Epomops franqueti*). In each map, the locations of reported observations of each species, extracted and curated from the Global Biodiversity Information Facility (***GBIF, 2014***) and used to train each model are given as grey points (*H. monstrosus*, n = 67; *E. franqueti*, n = 120 and *M. torquata*, n = 52). Expert opinion maps of the known range of each species, generated by the IUCN (***Schipper et al., 2008***), are outlined in grey. The colour legend represents a scale of the relative probability that the species occurs in that location from 0 (white, low) to 1 (green, high). Area under the curve statistics, calculated under a stringent ten-fold cross validation procedure, are 0.63 ± 0.04, 0.59 ± 0.04 and 0.58 ± 0.03 for *H. monstrosus*, *M. torquata* and *E. franqueti* respectively. (**D**) Is a composite distribution map giving the mean, relative probability of occurrence from (**A–C**).

transmission are predicted in a further 15 countries where, to date, index cases of the four African species of *Ebolavirus* have not been recorded. These are Nigeria, Cameroon, Central African Republic (CAR), Ghana, Liberia, Sierra Leone, Angola, Tanzania, Togo, Ethiopia, Mozambique, Burundi, Equatorial Guinea, Madagascar and Malawi (hereafter Set 2).

The AUC for the Ebola model was relatively high (AUC = 0.85 ± 0.04, n = 500 submodels) indicating that the model could strongly distinguish regions of environmental suitability for EVD. Enhanced vegetation index had the greatest impact on the distribution (relative contribution [RC] of 65.3%) followed by elevation (RC = 11.7%), night-time land surface temperature (LST) (RC = 7.7%), potential evapotranspiration (PET) (RC = 5.7%) and combined bat distribution (RC = 3.8%). Marginal effect plots are presented in *Figure 5—figure supplement 2*.

In total, 22.2 million people are predicted to live in areas suitable for zoonotic transmission of Ebola. The vast majority, 21.7 million (approximately 97%), live in rural areas, as opposed to urban or peri-urban areas (***CIESIN/IFPRI/WB/CIAT, 2007***; ***WorldPop, 2014***). Of these, 15.2 million are in Set 1 and 7 million are in Set 2. In terms of ranked populations at risk, DRC, Guinea and Uganda are highest

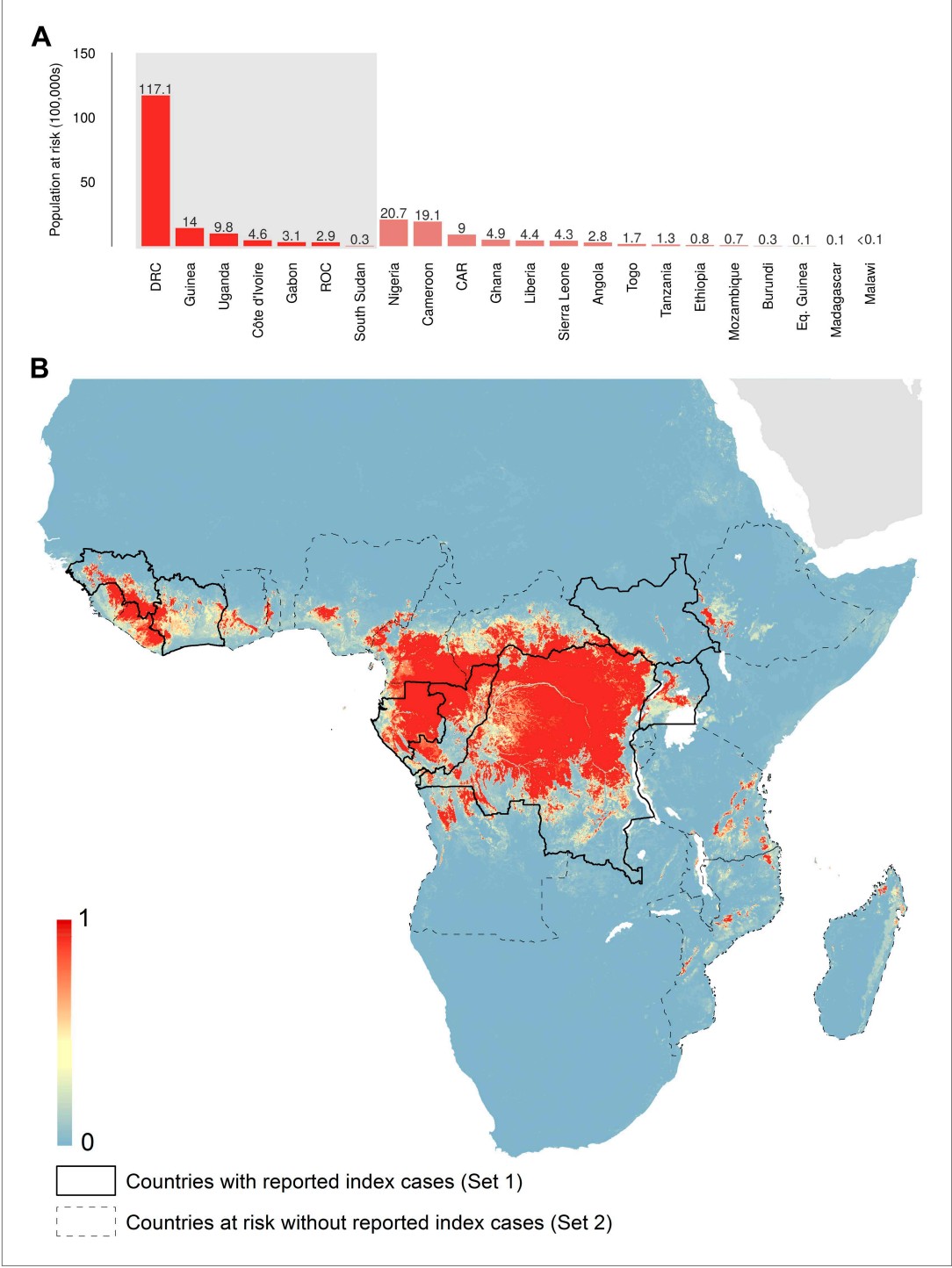

**Figure 5**. Predicted geographical distribution of the zoonotic niche for Ebola virus. (**A**) Shows the total populations living in areas of risk of zoonotic transmission for each at-risk country. The grey rectangle highlights countries in which index cases of Ebola virus disease have been reported (Set 1); the remainder are countries in which risk of zoonotic transmission is predicted, but in which index cases of Ebola have not been reported (Set 2). These countries are ranked by population at risk within each set. The population at risk Figure in 100,000 s is given above each bar. (**B**) Shows the predicted distribution of zoonotic Ebola virus. The scale reflects the relative probability that zoonotic transmission of Ebola virus could occur at these locations; areas closer to 1 (red) are more likely to harbour zoonotic transmission than those closer to 0 (blue). Countries with borders outlined are those which are predicted to contain at-risk areas for zoonotic transmission based on a thresholding approach (see 'Materials and methods').
*Figure 5. Continued on next page*

*Figure 5. Continued*

The area under the curve statistic, calculated under a stringent 10-fold cross-validation procedure is 0.85 ± 0.04. Solid lines represent Set 1 whilst dashed lines delimit Set 2. Areas covered by major lakes have been masked white.

The following figure supplements are available for figure 5:

**Figure supplement 1**. Covariates used in predicting zoonotic transmission niche of Ebola.

**Figure supplement 2**. Marginal effect plots for each covariate used in the Ebola virus distribution model.

**Figure supplement 3**. Comparison of predictions for zoonotic niche of Ebola virus excluding the Guinea outbreak.

---

in Set 1 and Nigeria, Cameroon and CAR are top in Set 2. For a full listing of these populations living in areas of risk, see the stacked bar plot in *Figure 5A*.

## National level demographic and mobility changes

Over the 40 year period since discovery of EVD, the total population living in those countries predicted to be within the zoonotic niche has nearly tripled (from 230 million to 639 million) and the proportion of the population in these countries living in an urban (rather than rural) setting has changed from 25.5% to 59.2% (*Figure 6*).

Data on the connectivity of human populations over this period were not available. We can infer however, intuitively, empirically and theoretically (*Zipf, 1946*; *Simini et al., 2012*) that rates of population movement within a country will scale directly in proportion to population growth.

## International connectivity by airline traffic

Records of passenger seat capacity are available since 2000 and show substantive increases over the period in Set 1 (from 2.96 to 4.77 million, a fractional change of 1.61) and Set 2 (from 5.6 to 15.6 million, a change of 2.8) (*Figure 7A*). More specific data on passenger volumes show almost universally similar increases since 2005 with Set 1 nations changing from 2 million to 2.5 million, a fractional change of 1.22 and Set 2 changing from 5 million to 7.9 million, a change of 1.57 (*Figure 7B*).

Global analysis of airline passenger volumes demonstrates that international connectivity has increased amongst all global regions and national income strata (*Figure 8*). Total passenger volumes

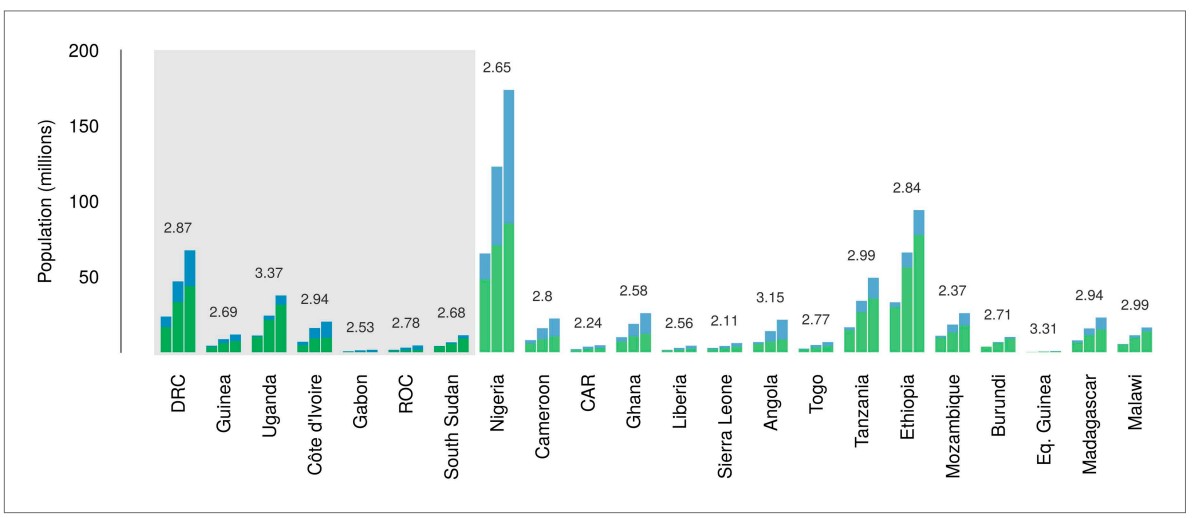

**Figure 6**. Changes in national population for countries predicted to contain areas at-risk of zoonotic Ebola virus transmission. For each country the population (in millions) is presented for three time periods (1976, 2000 and 2014) as three bars. Each stacked bar gives the rural (green) and urban (blue) populations of the country. The grey rectangle highlights countries in which index cases of Ebola virus diseases have been reported (Set 1); the remainder are countries in which risk of zoonotic transmission is predicted, but where index cases have not been reported (Set 2). The fractional change in population between 1976 and 2014 is given above each set of bars.

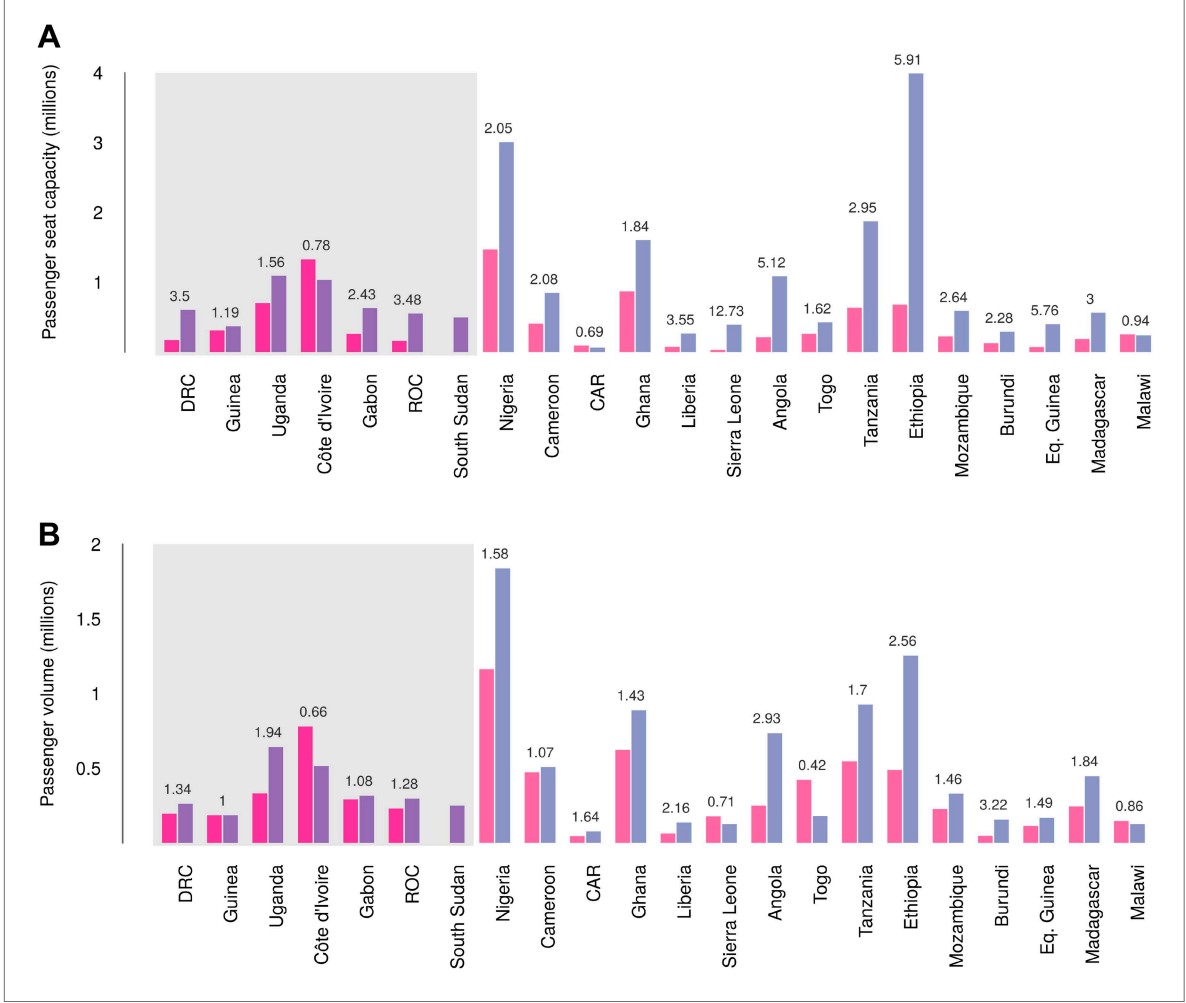

**Figure 7**. Changes in international flight capacity and traveller volumes for countries predicted to contain areas at-risk of zoonotic Ebola virus transmission. The grey rectangle highlights countries in which index cases of EVD have been reported (Set 1). The remainder are countries in which risk of zoonotic transmission is predicted, but where index cases have not been reported (Set 2). (**A**) Shows changes in annual outbound international seat capacity (between 2000 in red and 2013 in blue). (**B**) Depicts changes in annual outbound international passenger volume by country (between 2005 in red and 2012 in blue). For each country, the fractional change in volume is given above each set of bars. Note that only one bar is presented for South Sudan as data for this region prior to formation of the country in 2011 were unavailable.

have increased by a third from 9.5 to over 14 million during the eight year window (2005–2012) where records are available. The largest increases have occurred in WHO regions (***WHO, 2014b***) outside of the sub-Saharan African region (AFRO) (***Figure 8A,B***). In 2012, almost half of the final destinations of those travelling from these at-risk countries were to other AFRO nations (47%). Other frequent destinations were in Europe (EURO; 27%) and the Eastern Mediterranean (EMRO; 13%). Similarly, analysis of passenger volumes by World Bank national income groupings (***WHO, 2014b***) (***Figure 8C,D***) show that in 2012 40% of all passenger final destinations were to low or low-middle income countries.

## Discussion

### Summary of the main findings

We have re-evaluated the zoonotic niche for EVD in Africa. In doing so we have (i) used all existing outbreaks to assemble an inventory of index cases (n = 30); (ii) added to this all confirmed records of Ebola virus in animals (n = 51); (iii) assembled more accurate and contemporary environmental covariates including new maps of the distribution of confirmed OWFB reservoirs of the disease; and (iv) used

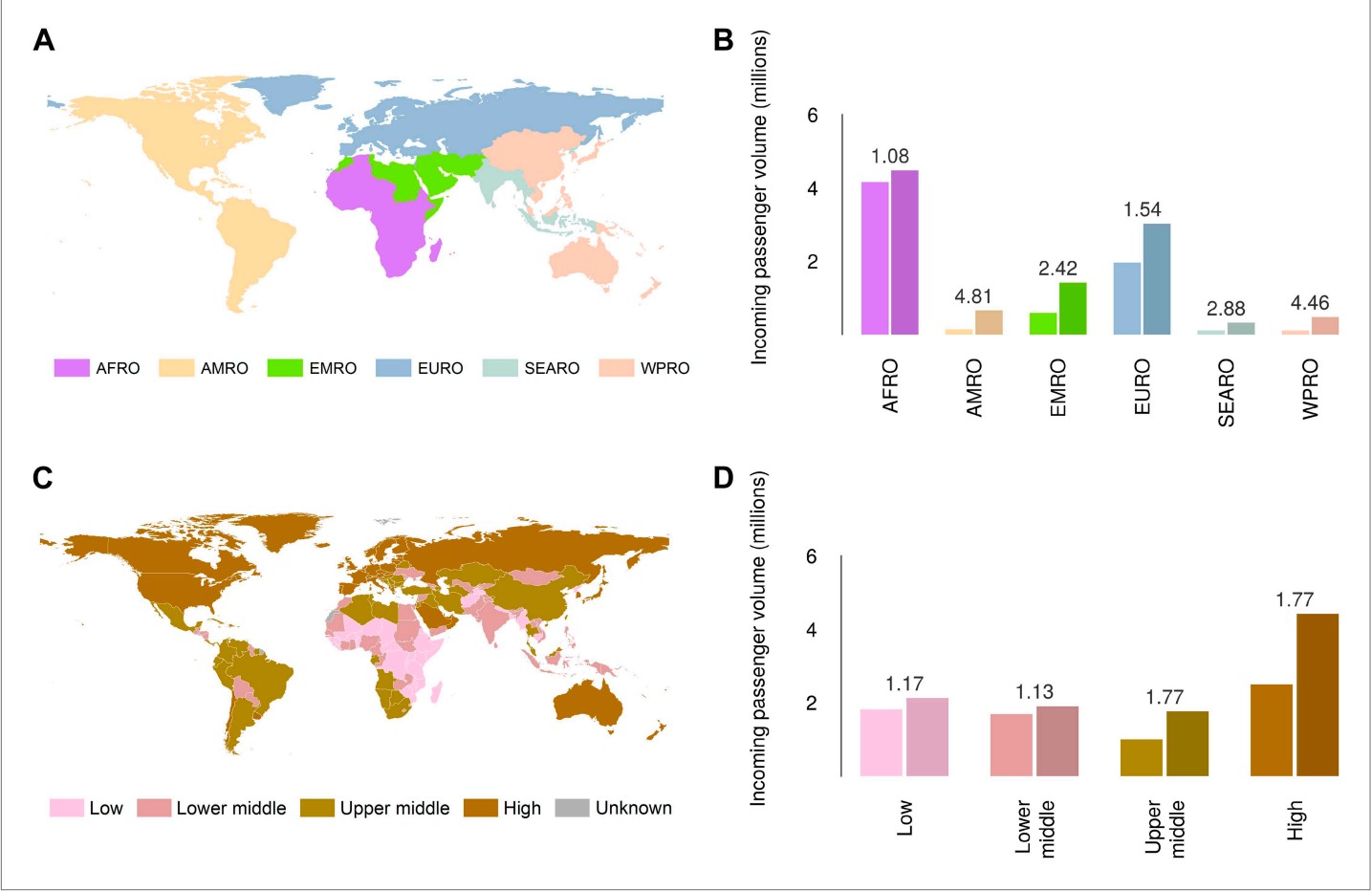

**Figure 8**. Numbers of airline passengers arriving from at-risk countries to other countries stratified by major geographic regions and national income groups. (**A**) Shows the locations of WHO regions (AFRO–African Region; AMRO–Region of the Americas; EMRO–Eastern Mediterranean Region; EURO–European Region; SEARO–South-East Asian Region; WPRO–Western Pacific Region). (**B**) Displays the numbers of passengers arriving in each of these regions from countries predicted to contain areas at risk of zoonotic Ebola virus transmission (Sets 1 and 2) in 2005 and 2012. (**C**) Shows the income tiers of all countries as defined by the World Bank. (**D**) Displays the total numbers of passengers arriving in countries in each of these income strata from at-risk countries in 2005 and 2012. The number above each pair of bars indicates the fractional change in these numbers of incoming passengers between 2005 and 2012.

the latest niche modelling techniques to predict the geographic distribution of potential zoonotic transmission of the disease. Using these predictions we have estimated the populations at risk of EVD both in countries which have confirmed index cases (Set 1, n = 7) and those for which we predict strong environmental suitability for outbreaks (Set 2, n = 15). In all countries at risk we show that since the discovery of EVD in 1976, urban and rural populations have increased and have become more interconnected both within and across national borders. During the last 40 years the increasing size and connectivity of these populations may have facilitated the subsequent spread of EVD outbreaks. These factors underline a change in the way in which EVD interacts with human populations.

## Interpreting the zoonotic niche

The remote and isolated nature of Ebola zoonotic transmission events, paired with the relatively poor diagnostics and understanding of the disease transmission routes in early outbreaks, mean that under-reporting of previous outbreaks is probable. An increasing understanding and description of a broader range of symptoms used in case definitions of EVD (*Leroy et al., 2000*; *Feldmann and Geisbert, 2011*) also increase the possibility that past outbreaks may have been misattributed to different diseases (*Tignor et al., 1993*). This poor detectability of EVD also clearly limits capacity to accurately

identify the locations and transmission routes of index cases (*Heymann et al., 1980*; *Baize et al., 2014*). We must assume, as has been done previously (*Peterson et al., 2004a*; *Jones et al., 2008*), that the first reported cases are representative of the true location of the index cases. Where possible we have represented this geographic uncertainty by attributing the index case to a wide-area polygon which then incorporated this uncertainty into the mapping process (see 'Materials and methods').

The relationship between the EVD niche and the environmental covariates (*Figure 5—figure supplement 2*), particularly the high relative contribution of the vegetation index, underscore that there are clear environmental limits to transmission of the virus from animals to humans, and that ecoregions dominated by rainforest are the primary home of such zoonotic cycles. Our analysis has shown that the zoonotic niche of the pathogen is more widespread than previously predicted or appreciated (*Peterson et al., 2004a*), most notably in West Africa.

This analysis used information from all human outbreaks and animal infections to delineate the likely zoonotic niche of the disease. Further analysis, excluding the existing outbreak focussed in Guinea from the dataset used to train the model (*Figure 5—figure supplement 3*), still resulted in prediction of high suitability in this region, with the presumed index village located within 5 km of an at-risk pixel. This implies that the eco-epidemiological situation in Guinea is very similar to that in past outbreaks, mirroring phylogenetic similarity in the causative viruses (*Dudas and Rambaut, 2014*; *Gire et al., 2014*). The ecological similarity between the past and current outbreaks also lends support to the notion that the scale of this outbreak is more heavily influenced by patterns of human-to-human transmission than any expansion of the zoonotic niche.

## Interpreting population at risk

It is important to appreciate that this zoonotic niche map delineates areas in which populations are at-risk of zoonotic transmission of EVD (*Figure 5B*). It does not predict the likelihood of EVD spill-over, the likelihood of an outbreak establishing, or its subsequent rate of spread within a population. Increasing human encroachment and certain cultural practices sometimes linked with poverty, such as bushmeat hunting, result in increasing exposure of humans to animals which may harbour diseases including Ebola (*Daszak, 2000*; *Wolfe et al., 2005*, *2007*). Increasing human population may accelerate the degree of risk through these processes but spatially refined information on these factors is not available comprehensively. It is hoped that as the understanding of the risk factors for zoonotic transmission of *Ebolavirus* to humans increases, it will be possible to incorporate this information into future risk mapping assessments.

Previous considerations of the geographic distribution of EVD have used human outbreaks alone. We have updated this work to include the last decade of outbreaks, as well as disaggregated outbreaks where evidence suggests multiple independent zoonotic transmission events overlap in space and time. Furthermore, our modelling process accommodates uncertainty in geopositioning of these index cases by utilising both point and polygon data. In addition, we include occurrence of infection in wildlife, important to the wider scale of zoonotic transmission (*Figure 1*), which in total has increased the dataset used in the model to 81 occurrences. The rareness of EVD outbreaks and the prevalence of detectable Ebola virus in reservoir species suggests that there will always be a limited set of observation data when compared to mapping of more prevalent zoonoses (*Pigott et al., 2014*). The results demonstrate predictive skill using a stringent validation procedure, however, indicating strong model performance even with this relatively limited observation dataset.

A broad zoonotic niche is predicted across 22 countries in Central and West Africa. Whilst several of these countries have reported index cases of EVD, others have not, although serological evidence in some regions points to possible underreporting of small-scale outbreaks (*Kuhn, 2008*). With improved ecological understanding, particularly with improvements to our knowledge of specific reservoir species and their distributions, it may be possible to delineate areas not at risk due to the absence of these species.

Despite relatively a large population living in areas of risk and the widespread practice of bushmeat hunting in these predicted areas (*Wolfe et al., 2005*; *Mfunda and Røskaft, 2010*; *Brashares et al., 2011*; *Kamins et al., 2011*), *Ebolavirus* is rare both in suspected animal reservoirs (*Leroy et al., 2005*; *Olival and Hayman, 2014*) and in terms of human outbreaks (*Table 1*). There is some indication however, that the frequency of Ebola outbreaks has increased since 2000, as shown in *Figure 2A*. We have shown that the human population living within this niche is larger, more mobile and better internationally connected than when the pathogen was first observed. As a result, when spillover events do

occur, the likelihood of continued spread amongst the human population is greater, particularly in areas with poor healthcare infrastructure (*Briand et al., 2014*; *Fauci, 2014*).

Whilst rare in comparison to other high burden diseases prevalent in this region, such as malaria (*Gething et al., 2011*; *Murray et al., 2012*), Ebola outbreaks can have a considerable economic and political impact, and the subsequent destabilisation of basic health care provisioning in affected regions increases the toll of unrecorded morbidity and mortality of more common infectious diseases (*Murray et al., 2014*; *Wang et al., 2014*), throughout and after the epidemic period. The number of concurrent infections during the present outbreak represents a significant strain on healthcare systems that are already poorly provisioned (*Briand et al., 2014*; *Chan, 2014*; *Fauci, 2014*) and many other Set 1 and Set 2 countries rank amongst the lowest per capita healthcare spenders. These considerations should be paramount when international organizations debate the financing requirements for the improvement of healthcare needed in the region and the urgency with which new therapeutics and vaccines can be brought into production (*Brady et al., 2014*; *Goodman, 2014*).

Together, these considerations necessitate prioritisation of efforts to reinforce and improve existing surveillance and control, and encourage the development of therapeutics and vaccines. The national population at risk estimates presented here would be a strong rationale for improving, prioritising and stratifying surveillance for EVD outbreaks and diagnostic capacity in these countries. We believe it would be prudent to test OWFB species in Set 2 countries for Ebola virus (*Hayman et al., 2012*), particularly during the bat breeding season to maximise chances of isolation in order to clarify the outbreak risk in these countries. In all Set 1 and Set 2 countries, raising awareness about the risk presented by reservoir bats and incidental primate hosts and the modes of transmission of this disease could be of value. Finally, increasing our capacity to rapidly map ever changing biological threats is also a core need (*Hay et al., 2013b*).

## Interpreting International connectivity

The increasing connectedness of the Africa region means that EVD is now a problem of international concern. While most EVD secondary transmission occurs locally and is likely transported via ground transit (*Francesconi et al., 2003*), the potential for international spread of infection is possible, as demonstrated by the importation of the disease from Liberia to Nigeria, culminating in further secondary transmission in Lagos (*WHO, 2014a*). The aetiology of EVD infection and disease progression means that an international outbreak propagated by air travel remains unlikely, particularly in high-income countries better able to handle EVD cases (*Fauci, 2014*). Nevertheless, a non-negligible threat remains, particularly in the low and middle income destinations and the rapid increase in global connectivity of these at-risk regions indicates that international airports could see more imported cases (*Chan, 2014*).

## Future work

We have focussed on reanalysing the zoonotic niche for EVD transmission and the characterisation of the populations at risk to improve the landscape in which future risk and impact of EVD outbreaks can be discussed. During the current emergency much of the work will concentrate on routes of secondary transmission in the human population—conceptually the red arrow of the H box in *Figure 1*. An important task is to stratify the risk of EVD spread both within and between countries and identify the most likely pathways of spread for characterisation and surveillance. Our next priority therefore is to investigate aspects of secondary human-to-human transmission by documenting the rate of geographic spread of EVD during the past and ongoing epidemics to help understand changes in these patterns in the historical record. Simulating these movements in a real landscape of population movement patterns, inferred from population movements assessed by mobile phones and other data (*Garcia et al., 2014*), as well as parametric movement models (*Simini et al., 2013*) is a logical next step, and can be used in future targeting of interventions and potential new treatments for both the current and future outbreaks (*Brady et al., 2014*; *Goodman, 2014*).

As previously discussed, whilst there is the risk of human travel during the latent phase of infection, and therefore potential for international spread, the high pathogenicity during infectiousness (immobilising infected persons) and the likely rapid and effective isolation measures implemented in regions with strong health care systems, limit the pandemic potential of EVD. Nevertheless, improvement of international containment plans and informed discussions of potential risks to airline carriers and populations of other regions will be supported by knowledge of local, regional and international population flows. Assessing these flows by air traffic volumes is an ongoing priority.

There are several other zoonotic viral haemorrhagic fevers (for example *Marburgvirus*, Lassa fever, hantaviral infections and arenaviruses) that are of similar public health and biosecurity concern (*Bannister, 2010*), due to their high virulence and mortality and their potential to cause outbreaks and spread internationally. Despite this their geographical distributions are poorly understood (*Hay et al., 2013a*). Many of the methods applied here can be adapted to these diseases and improve our geographical understanding of the risk presented by these pathogens.

We are in the midst of a public health emergency that will likely last for many more months (*Chan, 2014*) and which has brought EVD to global attention. We emphasise that the maps of zoonotic transmission presented here do not enable assessment of secondary transmission rates in human populations, but they do act as an evidenced-based indicator of locations with potential for future zoonotic transmission and thus outbreaks. Interestingly, early reports of another independent zoonotic outbreak in the DRC (*MSF, 2014*) are in predicted at-risk areas. An improved understanding of the spatial extent of the zoonotic niche can only help future efforts in biosurveillance.

## Materials and methods

### Methodological overview

A boosted regression tree (BRT) modelling framework was used to generate predictive risk maps of the zoonotic Ebola virus niche in Africa. This methodology combines regression trees, where trees are built according to optimal decision rules based on how binary decisions best accommodate a given dataset (*De'ath, 2007*; *Elith et al., 2008*), and boosting, which selects the tree that minimises the loss function. In doing so, a parameter space is defined which captures the greatest amount of variation present in the dataset. In order to train the model, four component datasets were compiled: (i) a comprehensive dataset of the reported locations of Ebola virus transmission from a zoonotic reservoir to a human; (ii) a dataset of the locations of Ebola virus infections in suspected reservoir and (non-human) susceptible host species (iii) a suite of ecologically relevant environmental covariates for Africa, including predicted distribution maps of suspected reservoir bat species and (iv) background (or pseudo-absence) records representing locations where zoonotic Ebola virus has not been reported. This study was limited to the African continent since no natural outbreaks of EVD have occurred outside the continent (*CDC, 2014*). Only *Reston ebolavirus* has a distribution reported outside of Africa, focussed in the Philippines, but has never been reported as pathogenic in humans; as a result this species was not included in the analysis.

### Identifying index cases and reconstructing zoonotic transmission events in space and time

Tables detailing proven outbreaks of Ebola virus, initially sourced from the scientific literature (*Kuhn, 2008*) and from health reporting organisations (*CDC, 2014*), were used to coordinate searches of the formal scientific literature using Web of Science and PubMed for each specific outbreak. Relevant papers were abstracted and where possible outbreak-specific epidemiological surveys were sourced. The citations in these references were obtained in order to reconstruct the outbreak in detail and to identify the most probable index case. Index cases were defined as any human infection resulting from interaction with non-human sources of the disease. Some of these cases arose from presumed interactions with zoonotic reservoirs or hosts, such as primates and other mammals during hunting trips (*Boumandouki et al., 2005*; *Nkoghe et al., 2005*, *2011*; WHO, 2003) or butchering of bats (*Leroy et al., 2009*). Any cases arising from existing human infections are considered as secondary infections rather than index cases. Similar to methodology employed elsewhere (*Messina et al., 2014*), the site, or supposed site, of this zoonotic transmission event was geopositioned using Google Earth. For locations where precise geographic information (e.g., geographic coordinates of a hunting camp) was provided by the authors, these were used. Where precise geographic information could not be accurately geopositioned, a geographic area (termed a polygon) was defined covering the reported region. In several cases only the first reported patient could be identified, with the source of infection unknown. With these outbreaks the location of the first patient was geopositioned under the assumption that an initial zoonotic spillover event occurred in the vicinity of this location. In two outbreaks multiple independent zoonotic transmission events were identified (WHO, 2003; *Nkoghe et al., 2005*; *Pourrut et al., 2005*), and each unique event was geopositioned and included in the model as separate entities. *Table 1* catalogues the outbreaks used in this study.

## Assembling a database of reported infections in animals

A literature search was conducted in Web of Science using the search term "Ebola" that returned 8635 citations. The abstracts were examined and for those that contained possible data on animal Ebola infection, the full text was obtained. The sampling site or location of the animal in the study was identified and geopositioned using Google Maps. These locations were recorded either as precise locations or as polygons, as with human index cases. Records for which local transmission of Ebola virus was deemed unlikely (e.g., seropositive primates tested in containment facilities several years after their capture) were excluded from the study. The non-human Ebola virus occurrence data collected are detailed in *Table 2*, including the diagnostic methods used.

## GenBank isolates

The open access sequence database GenBank (*NCBI, 2014*) was searched using MESH Umbrella search terms for Ebola virus, returning 181 results. These were then manually cross-referenced with the existing human and animal Ebola information, collected above, and 30 duplicates were removed. For the remaining isolates, original references and GenBank information fields were examined, but as there was insufficient information to establish precise location of isolation and/or whether the isolate represented an index case for any of these data sources, they were excluded from subsequent analyses.

## Covariates assembled and used in the analyses

A suite of ecologically relevant gridded environmental covariates for Africa was compiled, each having a nominal resolution of 5 km × 5 km. The environmental covariates used in this analysis were: elevation (from the shuttle radar topography mission [*ORNL DAAC, 2000*]); the mean value, and a measure of spatial variation (range, described below) between 2000 and 2012 of Enhanced Vegetation Index (EVI), daytime Land Surface Temperature (LST) and night-time LST; and mean potential evapotranspiration from 1950–2000 (*Trabucco and Zomer, 2009*) (*Figure 5—figure supplement 1*).

The EVI and LST datasets were derived from satellite imagery collected by NASA's Moderate Resolution Imaging Spectroradiometer (MODIS) remote sensing platform (*Tatem et al., 2004*). EVI is a metric designed to characterise vegetation density and vigour based on the ratio of absorbed photosynthetically active radiation to near infrared radiation (*Huete et al., 2002*). LST is a modelled product derived from emissivity as measured by the MODIS thermal sensor (*Wan and Li, 1997*), which is correlated, though not synonymous with air temperature, and effective for differentiating landscapes based on a combination of thermal energy and properties of surface types. The MODIS datasets utilized in this research (EVI was derived from the MCD43B4 product and the MOD11A2 LST product was used directly) were acquired as composite datasets created using imagery collected over multiple days, a procedure that results in products with 8-day temporal resolutions. Despite compositing, the EVI and LST datasets contained gaps due to persistent cloud cover found in forested equatorial regions, and these gaps were filled using a previously described approach (*Weiss et al., 2014a*). The EVI and LST datasets were then aggregated from their native 1 km × 1 km spatial resolution to a final 5 km × 5 km resolution, calculating both the mean and the range of the values of the subpixels making up each larger pixel. These spatial summaries therefore characterise both the mean temperature in each location as well as the degree of spatial heterogeneity within that pixel. This is of interest as humans and susceptible species are more likely to come into contact in transitional areas (e.g., boundary areas between areas of highly suitable susceptible species habitat and areas heavily utilised by humans). The final covariate production step consisted of summarising temporally across the 13-year data archive to produce synoptic datasets devoid of annual or seasonal anomalies (*Weiss et al., 2014a*).

## Implicated bat reservoir distributions

Over recent years, significant research has been undertaken in investigating the role bats have to play in the transmission cycle of Ebola viruses (*Olival and Hayman, 2014*) and evidence of asymptomatic infection in fruit bats has been documented to varying extents (*Leroy et al., 2005*; *Pourrut et al., 2007*, *2009*; *Hayman et al., 2010*; *Hayman et al., 2012*). In order to incorporate this potential driver of Ebola virus transmission into the model we developed predicted distribution maps for three species of fruit bat implicated as primary reservoirs of the disease: *Hypsignathus monstrosus*, *Epomops franqueti* and *Myonycteris torquata*. The evidence was strongest for these three species having a reservoir role

as Ebola virus RNA (all nested within the *Zaire ebolavirus* phylogeny [*Leroy et al., 2005*]) has been detected in all three. Whilst a handful of other bat species have been found to be seropositive, no further viral isolations have been recorded (*Olival and Hayman, 2014*).

Whilst expert opinion range maps for these species exist (*Schipper et al., 2008*), there is some disagreement with independently-sourced occurrence data (all archived in the Global Biodiversity Information Facility). As a result, a predictive modelling approach was used to create a continuous surface of habitat suitability for these species which we then included as a predictor in the model. Occurrence data for all Megachiroptera in Africa was extracted from GBIF (*GBIF, 2014*) using the R packages dismo (*Hijmans et al., 2014*) and taxize (*Chamberlain et al., 2014*). To remove apparently erroneous records in the GBIF archive all occurrence records more than 100 km from the species known ranges, as determined by expert-opinion range maps (*Schipper et al., 2008*), were excluded, as were duplicate records and those located in the ocean. This resulted in a total dataset of 1341 unique occurrence records.

The occurrence database was then used to train separate boosted regression tree species distribution models (*Elith et al., 2008*) to predict the likely distribution of each of these suspected reservoir species. For each model, occurrence records for the target species (*H. monstrosus*, n = 67; *E. franqueti*, n = 120; and *M. torquata*, n = 52) were considered presence records and occurrence records of all other species were used as background records. This procedure is designed to account for the potentially biasing effect of spatial variation in recording of Megachiroptera occurrences (*Phillips et al., 2009*).

For each species we ran fifty submodels each trained to a randomly selected bootstrap of this dataset, subject to the constraint that each bootstrap contained a minimum of 10 occurrence and 10 background records. Each submodel was fitted using the gbm.step subroutine (*Elith et al., 2008*) in the dismo R package. In each submodel the background records were down weighted so that the weighted sum of presence records equalled the weighted sum of background records (*Barbet-Massin et al., 2012*) in order to maximise the discrimination capacity of the model. We generated a prediction map from each of these submodels and calculated both the mean prediction and 95% confidence interval around the prediction for each 5 km × 5 km pixel for each species.

Model accuracy was assessed by calculating the mean area under the curve (AUC) statistic for each submodel under a stringent 10-fold cross validation for each submodel and obtaining the mean and standard deviation across all 50 submodels. Under this procedure the dataset was split into ten subsets, each containing approximately equal numbers of presence and background points. The ability of a model trained on each subset to predict the distribution of the other 90% of records was assessed by AUC and the mean value taken. As so few presence records were used to train each fold model (i.e., around 5 presence records for *M. torquata* up to 12 for *E. franqueti*), this represents a very stringent test of the model's predictive capacity. Additionally, to prevent inflation of the accuracy statistics due to spatial sorting bias, these statistics were estimated using a pairwise distance sampling procedure (*Hijmans, 2012*). Consequently, the AUC statistics presented here are lower than would be returned by standard procedures but gives a more realistic quantification of the model's ability to extrapolate predictions to new regions (*Wenger and Olden, 2012*). We also generated marginal effect plots with associated uncertainty intervals and relative contribution statistics (how often each covariate was selected during the model fitting process) as quantification of the sensitivity of the model to the different covariates. These allow us to make inferences about the ecological relationship between each species and its environment as well as to identify where this relationship is most uncertain.

To generate a single surface describing the distribution of the bat reservoir species to be used as a covariate in the subsequent Ebola modelling, the three mean prediction distribution maps were merged by taking the average habitat suitability for each of the three bat species at each pixel.

## Ebola distribution modelling

The Ebola virus occurrence dataset was supplemented with a background record dataset generated by randomly sampling 10,000 locations across Africa, biased towards more populous areas as a proxy for reporting bias (*Phillips et al., 2009*). We fitted 500 submodels to bootstraps of this dataset. To account for uncertainty in the geographic location of those occurrences reported as polygons, for each submodel one point was randomly selected from each of these occurrence polygons. This Monte

Carlo procedure enabled the model to efficiently integrate over the environmental uncertainty associated with imprecise geographic data. A bootstrap sample was then taken from each of these datasets and used to train the BRT model using the same procedure and weighting of background records as for the bat distribution models. Similarly, we generated a prediction map from each of these models and calculated both the mean prediction and corresponding 95% confidence intervals for each pixel and analysed prediction accuracy using the same stringent cross validation and sensitivity analysis procedure as for the bat distribution models (detailed above).

The predicted distribution map produced by this approach represents the environmental suitability of each pixel for zoonotic Ebola virus transmission. This may be interpreted as a relative probability of presence in the sense that more suitable pixels are more likely to contain zoonotic transmission than less suitable pixels, though not an absolute probability that transmission occurs in a given pixel. Similarly, the presence of zoonotic transmission increases the risk of transmission to a human, though this is also contingent on how humans interact with these zoonotic pools, through hunting or other activities.

### Population living in areas of environmental suitability for zoonotic transmission

In order to identify areas which are likely to be at risk of transmission of *Ebolavirus* from zoonotic reservoir hosts to humans, the continuous map of the predicted environmental suitability for zoonotic transmission (shown in *Figure 5*) was converted into a binary map classifying pixels as either at risk or not at risk. A pixel was assumed to be at risk if its predicted environmental suitability for zoonotic Ebola virus transmission was greater than 0.673, the lowest suitability value predicted at the locations of known transmission to humans (point records of human index cases). Countries containing at least one at-risk pixel are shown in *Figure 5B*—those countries that previously report an index case were defined as Set 1; countries with at least one at-risk pixel with no previous index cases of EVD were categorised as Set 2. The number of people living in at-risk areas in each of these countries was calculated by summing the estimated population of at-risk pixels using population density maps from the AfriPop project (*Linard et al., 2012*; *WorldPop, 2014*) and the proportion of those living in urban, periurban and rural areas was evaluated using the Global Rural Urban Mapping Project (*CIESIN/IFPRI/WB/CIAT, 2007*).

The R code used for all of the analysis has been made available on an open source basis (https://github.com/SEEG-Oxford/ebola_zoonotic).

### National level demographic and mobility data

For three separate years (1976, 2000 and 2014), total national populations were retrieved and the proportion of rural to urban populations noted from World Bank statistics (*World Bank, 2014*). To describe global air travel patterns from Set 1 and Set 2 countries, flight schedules data from the Official Airline Guide, reflecting an estimated 95% of all commercial flights worldwide, were analysed between 2000 and 2013 to calculate the annual volume of seats on direct flights that depart from each predicted country and which have an international destination. Complementing these seat capacity data, worldwide data on anonymised, individual passenger flight itineraries from the International Air Transport Association (2012) (*IATA, 2014*) were analysed between 2005 and 2012 to calculate the annual volume of international passenger departures out of each Set 1 and Set 2 country. The IATA dataset represents an estimated 93% of the world's commercial air traffic at the passenger level and includes points of departure and arrival and final destination information for travellers as well as their connecting flights.

## Acknowledgements

We thank Katherine Battle, Maria Devine and Kirsten Duda for proof-reading and Jane Messina for creating *Figure 1*. We also thank Andrew Rambaut for his comments on the final draft.

## Additional information

### Competing interests

SIH: Reviewing editor, *eLife*. The other authors declare that no competing interests exist.

## Funding

| Funder | Grant reference number | Author |
|---|---|---|
| University Of Oxford | Sir Richard Southwood Graduate Scholarship | David M Pigott |
| Bill and Melinda Gates Foundation | OPP1053338 | Nick Golding |
| Medical Research Council | K00669X | Peter W Gething |
| Biotechnology and Biological Sciences Research Council | Studentship | Oliver J Brady |
| German Academic Exchange Service | Graduate Scholarship | Moritz UG Kraemer |
| U.S. National Library of Medicine | R01LM010812 | John S Brownstein, Sumiko R Mekaru |
| 7th Framework Programme for Research and Technological Development (EU FP7) | 602525 | Peter W Horby |
| Canadian Institutes of Health Research | | Isaac I Bogoch, Kamran Khan |
| Wellcome Trust | 095066 | Adrian Mylne, Simon I Hay |
| RAPIDD program of the Science & Technology Directorate | | Andrew J Tatem, Simon I Hay |
| Fogarty International Center | | Andrew J Tatem, Simon I Hay |
| Bill and Melinda Gates Foundation | OPP1106023 | Zhi Huang, Andrew J Henry, Catherine L Moyes |
| Bill and Melinda Gates Foundation | OPP1068048 | Daniel J Weiss, Samir Bhatt |
| Bill and Melinda Gates Foundation | OPP1093011 | Catherine L Moyes, John S Brownstein, Sumiko R Mekaru, Simon I Hay |
| National Institute of Allergy and Infectious Diseases | U19AI089674 | Andrew J Tatem |
| Bill and Melinda Gates Foundation | OPP1106427 | Andrew J Tatem |
| Bill and Melinda Gates Foundation | OPP1032350 | Andrew J Tatem |

The funders had no role in study design, data collection and interpretation, or the decision to submit the work for publication.

## Author contributions

DMP, Advised and assembled the outbreak and animal infection data, Wrote the article; NG, Extracted the bat data, Conducted all of the analysis, Drafting or revising the article; AM, Assembled and geo-positioned the outbreak and animal infection data, Drafting or revising the article; ZH, Geo-positioned the outbreak and animal infection data, Provided all maps, Drafting or revising the article; AJH, Analysed international flight data, Helped assemble maps, Drafting or revising the article; IIB, Assisted with international transportation analysis, Edited the final draft of the article; SRM, Assisted in geopositioning, Drafting or revising the article; AJT, Provided information on urban change and migration data, Drafting or revising the article; KK, Provided data, Conducted all analyses on international air traffic patterns, Drafting or revising the article; DJW, Assembled the covariate layers, Drafting or revising the article; SB, Extracted Ebola information from GenBank, Drafting or revising the article; OJB, Screened GenBank data, Provided *Figure 2A*, Drafting or revising the article; MUGK, DLS, CLM, Analysis and interpretation of data, Drafting or revising the article; PWG, Assembled the covariate layers, Drafting or revising the article; PWH, JSB, Advised on international public health context, Edited the final draft of the article; SIH, Conceived the work and analysis, Wrote content and edited the article at all stages of development, Acts as guarantor of the paper

## Author ORCIDs

David M Pigott, http://orcid.org/0000-0002-6731-4034
Nick Golding, http://orcid.org/0000-0001-8916-5570
David L Smith, http://orcid.org/0000-0003-4367-3849
Simon I Hay, http://orcid.org/0000-0002-0611-7272

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
