## [Decision Letter]

Thank you for sending your work entitled “Mapping the zoonotic niche of Ebola virus disease in Africa” for consideration at *eLife*. Your article has been favorably evaluated by Prabhat Jha (Senior editor), who is overseeing the peer review of your article, three members of our Board of Reviewing Editors, and we have also received comments from five experts from outside of the Board of Reviewing Editors. Two of the reviewers, Pasi Penttinen and Alimuddin Zumla, have agreed to reveal their identity.

The Senior editor has assembled the following comments to help you prepare a revised submission. We request the specific important revisions to ensure high scientific integrity of a very visible, and topical subject, given the on-going attention to Ebola virus.

1) The manuscript should focus mostly on Ebola risk, with less emphasis on other parts. (You can consider if primate risk might be dropped for now.)

2) The 22 million people at risk needs important caveats as most of these live in cities/towns and are at risk of secondary spread. We would like to see more analysis/discussion of secondary spread potential versus primary fruit bat exposure risk, which is the basis for the 22M.

3) The discussion of air travel-based risk seems incomplete: the risk of air travel-related transmission between Set 1 and Set 2 countries is not provided, but would appear to be crucial.

4) Similarly, one reviewer points out a possible role of rivers in disrupting primate transmission of other viruses. Are there similar geographic variables to account for EBV variation in risk?

5) There is not sensitivity analyses in the Methods section to outline to which of the variables the final risk estimates were most sensitive. If possible, some uncertainty estimates on the population at risk rather than a single point estimate would be helpful.

6) Similarly, the Discussion requires a clear discussion about the limitations of the methods (pointed out by reviewer 3 in particular, but other reviewers also).

7) The Abstract needs to provide greater detail about exposure risk versus secondary transmission risk.

8) The health system data/spending has recently been covered also in the NEJM. You might consider dropping this as it is not central to the argument, and it is already well known that health spending and capacities are low in the EBB-affected regions. These recent papers should be cited, however.

We encourage your team to carefully examine all the language in the Discussion and in the Abstract in particular to be as cautious as possible. We recognize urgency of the paper, and the difficulties working with limited data, but this places special emphasis on very cautious interpretation.

All the review comments now follow. Please pay particular attention to the key comments in italics.

Reviewer 1

This is a subject of global importance. Pigott et al have written an excellent, clear, and comprehensive review putting together all available and latest data on EVD on recorded zoonotic transmission to humans, other primates and bats and have coupled this other environmental information to model the zoonotic niche of EVD in Africa which has then been mapped.

If extra space allows, the following could be mentioned:

1) The concluding paragraphs could point out the global disparities in healthcare pointed out by Prof fauci in the NEJM last week (Ref: Fauci AS. Ebola: Underscoring the Global Disparities in Health Care Resources. N Engl J Med. 2014 Aug 13 [PMID: 25119491])

2) A brief mention of the effects of potential effects of EVD on the distribution of non-human primates. Inogwabini BI1, Leader-Williams N. Effects of epidemic diseases on the distribution of bonobos. PLOS ONE. 2012;7(12):e51112. doi: 10.1371/journal.pone.0051112: This study examined how outbreaks and the occurrence of Anthrax, Ebola, Monkeypox and Trypanosomiasis may differentially affect the distribution of bonobos (*Pan paniscus*). Using a combination of mapping, Jaccard overlapping coefficients and binary regressions, the study determined how each disease correlated with the extent of occurrence of, and the areas occupied by, bonobos. Ebola, Monkeypox and Trypanosomiasis were each reported within the area of occupancy of bonobos. *Their results suggest that large rivers may have prevented Ebola from spreading into the range of bonobos.*

Reviewer 2

This is a highly topical and ambitious study improving previously published predictions of environmental suitability of Ebola virus disease transmission to humans. It has been designed, analysed and written up by an experienced group of epidemiologists, modellers, and public health professionals.

There is a public health need to get such analyses published rapidly to aid the ongoing, unprecedented outbreak response in Western Africa. The paper is analysed, presented, and written clearly, and the conclusions are sound and based on the results of the analyses.

I have two main concerns regarding this paper:

1) *I do not understand how the information on travel patterns falls in the scope of this paper.* If the travel patterns between countries potentially at risk were used in the models, please clarify in methods descriptions. If they were not used, I suggest considering leaving this only in discussion, or perhaps developing a follow-up manuscript of the international risks. The same comment applies to the comparison of the health systems strengths.

2) *I struggle to understand the relationship with the animal distribution and the other environmental factors in the models.* Should one not consider the presence of a suspected vector as an “obligatory” factor necessary for transmission, while the other factors are “enabling” or “facilitating” factors, therefore the risk zones should include only zones with potential or confirmed host animal presence. It is not immediately obvious from the maps or the discussion. Please clarify in the Discussion.

Reviewer 3

1) As someone involved in both the research and operational aspects of medical intervention in West Africa, I find nothing at all new about what is stated in this paper. In essence, what the authors are remarking is that population growth (and demographic changes), mobility, and poor public health infrastructure are risk factors for EVD. The new (ish) component of this paper is that they have done a good job of synthesizing diverse datasets into a clear and understandable way to present this information for a general scientific audience.

2) *The language in the conclusion, as mentioned, needs to be toned way down.* Although I appreciate and understand the need to have a way forward with respect to research, for those of us involved in the response to this epidemic, clearly, everyone recognizes the need to strengthen surveillance in the region, and the authors need to be careful in terms of how the conclusion is framed so as not to imply that this is somehow not recognized. I would strongly suggest that they take another stab at the conclusion and also make reference to several of the very nice perspective pieces in the NEJM that have been ongoing.

3) It is interesting that the authors do not make great use of the environmental aspects. Perhaps I misread this part in the paper, but it would be important to emphasize that *increasing contact with potential reservoirs* (*whether due to poverty, culture or deforestation) plays a role*. The component on reservoir mapping and the vegetation index I believe touches on this, but perhaps I missed this aspect. This is also arguably the case with respect to food insecurity. For example, in Guinea**,**
*one of the major difficulties in implementing public awareness measures to prevent hunting and eating of bushmeat is the need to replace bushmeat with some other food!* The authors, unless they did and I missed it, would do well to discuss the interaction of diverse sectors in terms of the health care infrastructure.

Reviewer 4

The authors are an experienced modelling group who have developed a novel approach to modelling risk. The model uses multiple data feeds to model the risk of zoonotic infection with Ebola virus infection (EBV) and by extension the risk to people.

The model uses animal infection climactic, topographical, airline travel, and human population growth to predict geographic areas at risk of EBV.

The contribution is novel and valuable. It theoretically expands the number of countries at risk from those in which it has occurred to 22 where conditions are favourable for its occurrence. *It would have been interesting to retroactively apply the model to past human outbreaks.*

The paper is very well written, very topical, and the work important. Moreover there may be applicability to other viral hemorrhagic fevers and perhaps other zoonotic disease.

Reviewer 5

The approach seems relevant and is (to me) original. The authors have a good understanding of all the uncertainties that are involved, as well as of the historical and historical context in which the epidemics take place, in their estimates and the manuscript is well written. If I were an editor at *eLife* I would go for it.

Reviewer 6

This looks like an interesting paper with some imaginative use of the data (in a good way). Some immediate thoughts:

(i) It needs a technical appendix to explain exactly how the various models were constructed.

(ii) This is really pushing the data as far as it can go, which is okay since it is all the data we have – *but the limitations of this exercise are not really discussed thoroughly*.

Reviewer 7

A paper on the Ebola epidemic is current and warrants fast tracking. However this manuscript is not addressing the key challenges and issues adequately, and unless authors can address this I don't think the manuscript in its current form is acceptable. *The important issues relate to risk*, the ethics relating to experimental drugs, and containment efforts.

Reviewer 8

The Ebola emergency clearly warrants urgent attention. Geospatial risk modeling provides potentially valuable information, and this report is timely. For public health planning we need to consider secondary spread particularly and this needs more development in the argument.

Main concerns:

1) Reads more like a review, albeit a good one.

2) Suggest paper focuses specifically on Ebola risk. Some of the more peripheral arguments can be condensed.

3) Needs clearer explanation of the limitations: specifically assumes equivalent fruit bat susceptibility and virus transmissibility. Bat to human transmission is relatively rare after all, yet the bats and the humans are both numerous.

4) 22 million people may live in the potentially affected regions but most of them are in cities/towns and are at risk of secondary spread. I would like to see more analysis/discussion of secondary spread potential versus primary fruit bat exposure risk.

5) It would be useful to identify key assumptions in the modeling to which the predictions are sensitive yet uncertainty is high as immediate research priorities.

---

## [Author Response]

Many interesting points were raised and we hope that any concerns have been suitably addressed in this revision. There have been early reports of another outbreak of Ebola virus disease in the Democratic Republic of Congo apparently arising from an independent zoonotic transmission event. At this time there is insufficient information to include this new outbreak in the model as precise information for the index case has not been reported, however we make reference to this in the conclusion and highlight that the locations of reported cases are within an area predicted to be at risk of zoonotic transmission. Consequently, addition of this data point would be unlikely to affect the current model predictions.

Reviewer 1

*This is a subject of global importance. Pigott et al have written an excellent, clear, and comprehensive review putting together all available and latest data on EVD on recorded zoonotic transmission to humans, other primates and bats and have coupled this other environmental information to model the zoonotic niche of EVD in Africa which has then been mapped*.

We thank the reviewer for the positive feedback.

If extra space allows, the following could be mentioned:

*1) The concluding paragraphs could point out the global disparities in healthcare pointed out by Prof fauci in the NEJM last week* (*Ref: Fauci AS. Ebola: Underscoring the Global Disparities in Health Care Resources. N Engl J Med. 2014 Aug 13 [PMID: 25119491])*

We have incorporated this reference, as well as the other *NEJM* articles that were published subsequent to our initial submission. As a result, we have also dropped Figure 9 and its accompanying text in the article.

*2) A brief mention of the effects of potential effects of EVD on the distribution of non-human primates. Inogwabini BI1, Leader-Williams N. Effects of epidemic diseases on the distribution of bonobos. PLOS ONE. 2012;7*(*12):e51112.* doi: 10.1371/journal.pone.0051112:
*This study examined how outbreaks and the occurrence of Anthrax, Ebola, Monkeypox and Trypanosomiasis may differentially affect the distribution of bonobos* (*Pan paniscus). Using a combination of mapping, Jaccard overlapping coefficients and binary regressions, the study determined how each disease correlated with the extent of occurrence of, and the areas occupied by, bonobos. Ebola, Monkeypox and Trypanosomiasis were each reported within the area of occupancy of bonobos. Their results suggest that large rivers may have prevented Ebola from spreading into the range of bonobos*.

This research is primarily focused on the impact of zoonotic transmission to humans, and the environmental data and occurrences from animal populations were used to supplement our understanding of this distribution. The nature of human and bonobo disease interaction is unclear, and we only included in the modelling framework the three bat species for which the evidence for being reservoir species is greatest. Gorillas and chimpanzees are suspected dead-end hosts, and therefore these animals were not considered in the same manner. Since bats are not restricted by physical landmarks in the same way as primates, we did not include this aspect in our analysis. It is hoped that our work in addressing zoonotic transmission risk will aid future research by others into primate risk.

Reviewer 2

*This is a highly topical and ambitious study improving previously published predictions of environmental suitability of Ebola virus disease transmission to humans. It has been designed, analysed and written up by an experienced group of epidemiologists, modellers, and public health professionals*.

*There is a public health need to get such analyses published rapidly to aid the ongoing, unprecedented outbreak response in Western Africa. The paper is analysed, presented, and written clearly, and the conclusions are sound and based on the results of the analyses*.

We thank the reviewer for the generally positive reaction to the paper.

I have two main concerns regarding this paper:

*1) I do not understand how the information on travel patterns falls in the scope of this paper. If the travel patterns between countries potentially at risk were used in the models, please clarify in methods descriptions. If they were not used, I suggest considering leaving this only in discussion, or perhaps developing a follow-up manuscript of the international risks. The same comment applies to the comparison of the health systems strengths*.

Analysis of population growth, travel patterns and healthcare expenditure was presented to provide evidence of the change in demographic and socioeconomic circumstances of populations living within areas suitable for zoonotic Ebola virus transmission and provide a quantitative summary of factors related to disease emergence, previously only qualitatively discussed. It is not provisioned to provide specific information on the current epidemic but to help wider thinking of the changing likelihood of secondary transmission of any initial zoonotic transmission event.

To address this concern, echoed by the editor, we have significantly trimmed down the respective sections, removing Figure 9 and the section on healthcare expenditure, and condensing the international air travel from three paragraphs to one. We also cite the recent perspective pieces from the *NEJM* as requested.

*2) I struggle to understand the relationship with the animal distribution and the other environmental factors in the models. Should one not consider the presence of a suspected vector as an “obligatory” factor necessary for transmission, while the other factors are “enabling” or “facilitating” factors, therefore the risk zones should include only zones with potential or confirmed host animal presence. It is not immediately obvious from the maps or the discussion. Please clarify in the Discussion*.

We agree that the absence of reservoirs would mean an absence of the disease in humans. However, there is imperfect knowledge on which organisms are the true hosts of Ebola and even for those species that we strongly suspect of being reservoirs there is only poor distributional information available. This can be seen by comparing GBIF data to the expert opinion range maps of Figure 4. As a result, for the three bat species that we include, we represented their distribution as a continuous surface of habitat suitability rather than a binary presence or absence. We subsequently incorporated this information into the EVD model and threshold human risk based on known human cases instead. We have clarified this in the manuscript as follows:

(in the Methods) “Whilst expert opinion range maps for these species exist (111), there is some disagreement with independently-sourced occurrence data (all archived in the Global Biodiversity Information Facility). As a result, a predictive modelling approach was used to create a continuous surface of habitat suitability for these species which we then included as a predictor in the model.”

(in the Discussion)“A broad zoonotic niche is predicted across 22 countries in Central and West Africa. Whilst several of these countries have reported index cases of EVD, others have not, although serological evidence in some regions points to possible underreporting of small-scale outbreaks (68). With improved ecological understanding, particularly with improvements to our knowledge of specific reservoir species and their distributions, it may be possible to delineate areas not at risk due to the absence of these species.”

Reviewer 3

*1) As someone involved in both the research and operational aspects of medical intervention in West Africa, I find nothing at all new about what is stated in this paper. In essence, what the authors are remarking is that population growth* (*and demographic changes), mobility, and poor public health infrastructure are risk factors for EVD. The new* (*ish) component of this paper is that they have done a good job of synthesizing diverse datasets into a clear and understandable way to present this information for a general scientific audience*.

We thank the reviewer for the critique, but we respectfully disagree. One key improvement is the updated risk map incorporating not just the more recent human outbreaks over previous publications, but also expanding the consideration of the niche of EVD to include infections across the Ebola transmission cycle. In addition, we have provided an evidence base for the changing demographic and socioeconomic situations in both Set 1 and Set 2 countries and indicate that in the future there is an increased likelihood of secondary transmission (in the absence of intervention) from any primary zoonotic transmission event. We are unaware of a previous quantitative assessment of these changes and therefore believe this to be a novel contribution. We have however reduced the prominence of these sections, particularly in the Discussion and Abstract in response to the review comments and editorial guidance.

*2) The language in the conclusion, as mentioned, needs to be toned way down. Although I appreciate and understand the need to have a way forward with respect to research, for those of us involved in the response to this epidemic, clearly, everyone recognizes the need to strengthen surveillance in the region, and the authors need to be careful in terms of how the conclusion is framed so as not to imply that this is somehow not recognized. I would strongly suggest that they take another stab at the conclusion and also make reference to several of the very nice perspective pieces in the NEJM that have been ongoing*.

We have reread the concluding paragraphs and are trying to relate our statements to the particular concerns of the referee. We have now cited extensively the *NEJM* papers that were published after initial submission. We have tried in a variety of places to tone down specific sentences. For instance:

(in the Discussion) “The aetiology of EVD infection and disease progression means that an international outbreak propagated by air travel remains unlikely, particularly in high-income countries better able to handle EVD cases (30). Nevertheless, a non-negligible threat remains, particularly in the low and middle income destinations and the rapid increase in global connectivity of these at-risk regions indicates that international airports could see more imported cases (19).”

We ask during the resubmission evaluation, if these adjustments have not been sufficient and there is still concern, that the editors highlight specific sentences.

*3) It is interesting that the authors do not make great use of the environmental aspects. Perhaps I misread this part in the paper, but it would be important to emphasize that increasing contact with potential reservoirs* (*whether due to poverty, culture or deforestation) plays a role. The component on reservoir mapping and the vegetation index I believe touches on this, but perhaps I missed this aspect. This is also arguably the case with respect to food insecurity. For example, in Guinea****,***
*one of the major difficulties in implementing public awareness measures to prevent hunting and eating of bushmeat is the need to replace bushmeat with some other food! The authors, unless they did and I missed it, would do well to discuss the interaction of diverse sectors in terms of the health care infrastructure*.

In the previous version of the manuscript we have not focused in detail on environmental and socioeconomic factors influencing contact with infected reservoir species. In response to this comment, we have added the following:

(in the Discussion) “Increasing human encroachment and certain cultural practices sometimes linked with poverty, such as bushmeat hunting, result in increasing exposure of humans to animals which may harbour diseases including Ebola (24; 147; 148). Increasing human population may accelerate the degree of risk through these processes but spatially refined information on these factors is not available comprehensively. It is hoped that as the understanding of the risk factors for zoonotic transmission of *Ebolavirus* to humans increases, it will be possible to incorporate this information into future risk mapping assessments.”

Reviewer 4

*The authors are an experienced modelling group who have developed a novel approach to modelling risk. The model uses multiple data feeds to model the risk of zoonotic infection with Ebola virus infection* (*EBV) and by extension the risk to people*.

*The model uses animal infection climactic, topographical, airline travel, and human population growth to predict geographic areas at risk of EBV*.

*The contribution is novel and valuable. It theoretically expands the number of countries at risk from those in which it has occurred to 22 where conditions are favourable for its occurrence. It would have been interesting to retroactively apply the model to past human outbreaks*.

*The paper is very well written, very topical, and the work important. Moreover there may be applicability to other viral hemorrhagic fevers and perhaps other zoonotic disease*.

We thank the reviewer for their comments. Our primary goal here was to map the zoonotic niche by using data from all outbreaks. We have already provided some of the analysis requested by the reviewer, in particular running a model iteration with the most recent outbreak excluded. The model trained using index cases before 2013 predicted risk within Guinea and very close to the reported index case for the 2013 outbreak. Similarly, all previous index cases occur within or near to areas of predicted highest risk. We highlight the section specific to Guinea and have added statements to clarify.

(in the Discussion) “This analysis used information from all outbreaks to delineate the likely zoonotic niche of the disease. Further analysis, excluding the existing outbreak focussed in Guinea from the dataset used to train the model (prediction maps available on request), still resulted in prediction of high suitability in this region, with the presumed index village located within 5km of an at-risk pixel. This implies that the eco-epidemiological situation in Guinea is very similar to that in past outbreaks, mirroring phylogenetic similarity in the causative viruses (26; 42).”

Reviewer 5

*The approach seems relevant and is* (*to me) original. The authors have a good understanding of all the uncertainties that are involved, as well as of the historical and historical context in which the epidemics take place, in their estimates and the manuscript is well written. If I were an editor at* eLife *I would go for it*.

We thank the reviewer for their endorsement.

Reviewer 6

*This looks like an interesting paper with some imaginative use of the data* (*in a good way). Some immediate thoughts:*

(*i) It needs a technical appendix to explain exactly how the various models were constructed*.

We have followed previous feedback from our last submission to *eLife* to provide a summary of the methods within the manuscript rather than a technical appendix. We therefore reference previous work, including “A working guide to Boosted Regression Trees” (outlining the core methodology) and disease-specific applications of this approach (for dengue, leishmaniasis and malaria vectors). We have now also uploaded the source code for this analysis to Github and provided a link to this in the paper. If directed to do so by the editors, we can provide more details on specific sections of the paper.

(*ii) This is really pushing the data as far as it can go, which is okay since it is all the data we have – but the limitations of this exercise are not really discussed thoroughly*.

We have added the following section:

(in the Discussion) “Previous considerations of the geographic distribution of EVD have used human outbreaks alone. We have updated this work to include the last decade of outbreaks, as well as disaggregated outbreaks where evidence suggests multiple independent zoonotic transmission events overlap in space and time. Furthermore, our modelling process accommodates uncertainty in geopositioning of these index cases by utilising both point and polygon data. In addition, we include occurrence of infection in wildlife, important to the wider scale of zoonotic transmission (Figure 1), which in total has increased the dataset used in the model to 81 occurrences. The rareness of EVD outbreaks and the prevalence of detectable Ebola virus in reservoir species suggests that there will always be a limited set of observation data when compared to mapping of more prevalent zoonoses (105). The results demonstrate predictive skill using a stringent validation procedure, however, indicating strong model performance even with this relatively limited observation dataset.”

Reviewer 7

*A paper on the Ebola epidemic is current and warrants fast tracking. However this manuscript is not addressing the key challenges and issues adequately, and unless authors can address this I don't think the manuscript in its current form is acceptable. The important issues relate to risk, the ethics relating to experimental drugs, and containment efforts*.

We agree with the reviewer that all these features are important, but have to respectfully disagree that the issues raised are important within the context of this work. We explain throughout the article that this work concerns primary transmission events and not secondary outbreaks, and therefore experimental drugs and containment efforts are not immediately relevant. In order to better clarify this distinction, we have amended the text in the discussion and incorporated references to drug development within the text, tying this to secondary transmission modelling.

(in the Discussion) “Simulating these movements in a real landscape of population movement patterns, inferred from population movements assessed by mobile phones and other data (35), as well as parametric movement models (116) is a logical next step, and can be used in future targeting of interventions and potential new treatments in both the current and future outbreaks (12; 44).”

Reviewer 8

*The Ebola emergency clearly warrants urgent attention. Geospatial risk modeling provides potentially valuable information, and this report is timely. For public health planning we need to consider secondary spread particularly and this needs more development in the argument*.

We agree with the reviewer that modelling secondary spread is important, and it is currently the focus of further work. We believe that it is important to refine our understanding of the risk of zoonotic transmission and believe we have outlined this clearly and consistently throughout the manuscript, and not made any claims about direct spread modelling. Anything concerning secondary transmission is in relation to the changing nature of humans living in areas at-risk, but not directly related to the potential magnitude of a specific outbreak. We have strengthened various parts of the manuscript to reflect the differences in approach that primary transmission and secondary transmission necessitate as detailed in the response to Reviewer 7.

Main concerns:

*1) Reads more like a review, albeit a good one*.

*2) Suggest paper focuses specifically on Ebola risk. Some of the more peripheral arguments can be condensed*.

In response to this and editorial guidance, we have dropped Figure 9 and its related content and cited the recent *NEJM* articles as requested by other reviewers in its place. We have decided to keep the brief descriptions of growth and travel volumes as this adds a quantitative analysis of features that have only so far been considered qualitatively, but these sections have been trimmed down in size, particularly in the discussion.

*3) Needs clearer explanation of the limitations: specifically assumes equivalent fruit bat susceptibility and virus transmissibility. Bat to human transmission is relatively rare after all, yet the bats and the humans are both numerous*.

Please refer to comments for Reviewer 6 discussing the areas of uncertainty with the work. We have already indicated that the disease is very rare in both bats and humans, e.g., in the Discussion:

(in the Discussion) “Despite relatively large population living in areas of risk and the widespread practice of bushmeat hunting in these predicted areas (13; 64; 83; 147), Ebolavirus is rare both in suspected animal reservoirs (77; 96) and in terms of human outbreaks (Table 1).”

*4) 22 million people may live in the potentially affected regions but most of them are in cities/towns and are at risk of secondary spread. I would like to see more analysis/discussion of secondary spread potential versus primary fruit bat exposure risk*.

In response to comments by Reviewer 7 we have made some edits to the text. In addition, we have now stated the proportion of the at-risk population living in rural areas:

(in the Results) “The vast majority, 21.7 million (approximately 97%), live in rural areas, as opposed to urban or peri-urban areas (21; 150).”

(in the Methods) “The number of people living in at-risk areas in each of these countries was calculated by summing the estimated population of at-risk pixels using population density maps from the AfriPop project (79; 150) and the proportion of those living in urban, periurban and rural areas was evaluated using the Global Rural Urban Mapping Project (21).

*5) It would be useful to identify key assumptions in the modeling to which the predictions are sensitive yet uncertainty is high as immediate research priorities*.

Our methods section was unclear that the marginal effects plots and relative contributions represent uncertainty and covariate sensitivity in our model. We have edited text in the Methods and figure legend for Figure 5—figure supplement 2 to reflect this.

(in the Methods) “We also generated marginal effect plots with associated uncertainty intervals and relative contribution statistics (how often each covariate was selected during the model fitting process) as quantification of the sensitivity of the model to the different covariates. These allow us to make inferences about the ecological relationship between each species and its environment as well as to identify where this relationship is most uncertain.”

Response to the editorial guidance:

*1) The manuscript should focus mostly on Ebola risk, with less emphasis on other parts.* (*You can consider if primate risk might be dropped for now.)*

As outlined in various responses above, we have streamlined a number of sections, specifically those relating to healthcare expenditure and the discussion about international travel. We have however included the corresponding results paragraphs, as this form part of the key narrative that the changing populations living at risk have the potential to influence the dynamics of future outbreaks.

*2) The 22 million people at risk needs important caveats as most of these live in cities/towns and are at risk of secondary spread. We would like to see more analysis/discussion of secondary spread potential versus primary fruit bat exposure risk, which is the basis for the 22M*.

The majority of these populations are rural. Please see comments for Reviewer 8.

*3) The discussion of air travel-based risk seems incomplete: the risk of air travel-related transmission between Set 1 and Set 2 countries is not provided, but would appear to be crucial*.

The context for air travel risk was for a potential future outbreak (i.e., an index case in either Set 1 or Set 2 which then subsequently spread from human-to-human to another location, including another Set 1 or 2 country), hence the global summary statistics for air travel. However, we have reduced the prominence of this section within the discussion considerably.

4) Similarly, one reviewer points out a possible role of rivers in disrupting primate transmission of other viruses. Are there similar geographic variables to account for EBV variation in risk?

Please see comments for Reviewer 1.

*5) There is not sensitivity analyses in the Methods section to outline to which of the variables the final risk estimates were most sensitive. If possible, some uncertainty estimates on the population at risk rather than a single point estimate would be helpful*.

Please see comments for Reviewers 6, 7 and 8 – we have clarified how such estimates can be inferred from our model. It was not possible to calculate uncertainty for the population at risk estimates.

*6) Similarly, the Discussion requires a clear discussion about the limitations of the methods* (*pointed out by reviewer 3 in particular, but other reviewers also)*.

Please see comments for Reviewer 3.

*7) The Abstract needs to provide greater detail about exposure risk versus secondary transmission risk*.

We have amended the abstract to be more focussed on primary zoonotic transmission.

*8) The health system data/spending has recently been covered also in the NEJM. You might consider dropping this as it is not central to the argument, and it is already well known that health spending and capacities are low in the EBB-affected regions. These recent papers should be cited, however*.

We have dropped this section and have included references to the recent *NEJM* articles throughout.